# Minerva: Reinforcement Learning with Verifiable Rewards for Cyber Threat Intelligence LLMs

## Abstract

Cyber threat intelligence (CTI) analysts routinely convert noisy, unstructured security artifacts into standardized, automation-ready representations. Although large language models (LLMs) show promise for this task, existing approaches remain brittle when producing structured CTI outputs and have largely relied on supervised fine-tuning (SFT). In contrast, CTI standards and community-maintained resources define canonical identifiers and schemas that enable deterministic verification of model outputs. We leverage this structure to study reinforcement learning with verifiable rewards (RLVR) for CTI tasks. We introduce *Minerva*, a unified dataset and training pipeline spanning multiple CTI subtasks, each paired with task-specific verifiers that score structured outputs and identifier predictions. To address reward sparsity during rollout, we propose *MinervaRL*, an auxiliary self-training mechanism that generates additional verified trajectories and distills them back into the model. Averaged across four backbones and 12 CTI benchmarks, MinervaRL improves the mean score by 15.8 percentage points over the corresponding base models and by 4.3 points over GRPO.

## 1 Introduction

Cyber threat intelligence (CTI) often requires mapping free-text security information into standardized labels, scores, and structured records that downstream tools can use (Xu et al., 2024). In practice, analysts map unstructured inputs such as vulnerability descriptions, detection rules, and incident narratives to standardized frameworks and identifiers, including MITRE ATT&CK (Adversarial Tactics, Techniques, and Common Knowledge) for adversary behavior, Common Vulnerabilities and Exposures (CVE) identifiers, Common Weakness Enumeration (CWE) labels, and Common Vulnerability Scoring System (CVSS) vectors and scores (MITRE Corporation, 2026d; Byers et al., 2022; MITRE Corporation, 2026c; FIRST.org, Inc., 2019). These standards enable consistent reporting and large-scale exchange through formats and protocols such as Structured Threat Information Expression (STIX) and Trusted Automated Exchange of Intelligence Information (TAXII) (Strom et al., 2020; OASIS Cyber Threat Intelligence (CTI) Technical Committee, 2021a;b). Figure 1 illustrates this setting with a public vulnerability description mapped to canonical CWE and CVSS outputs. These standards also make correctness critical: a CTI system must ground outputs to evolving taxonomies, preserve evidence from noisy text, and produce syntactically valid structured artifacts. Even small identifier, schema, or formatting errors can break downstream automation or propagate incorrect intelligence.

Large language models (LLMs) are a natural fit for CTI because they can read long-form security narratives and generate structured analyst-facing outputs. Prior work shows growing use of LLMs across cybersecurity, and domain-adapted models such as CTI-BERT demonstrate that security-specific pretraining improves CTI-oriented extraction and representation learning (Xu et al., 2024; Park & You, 2023). However, existing CTI benchmarks reveal uneven reliability. Models can often follow analyst-style instructions and recover surface facts, but still fail on workflow-critical outputs such as ATT&CK technique mapping, mitigation recommendation, and vulnerability root-cause identification (Ji et al., 2024; Alam et al., 2024; Liu et al., 2024). These failures are especially problematic for deployment, where CTI systems must produce canonical identifiers, valid schemas, and grounded structured predictions, often under constraints that favor smaller specialized models.

> **Representative CTI mapping**
>
> **Input: CVE description**
> *CVE-2024-20092*. In vdec, there is a possible out of bounds write due to a missing bounds check. This could lead to local escalation of privilege with System execution ...
>
> **Structured output**
> CWE: `CWE-787`
> CVSS: `AV:L/AC:L/PR:L/UI:N/S:U/C:H/I:H/A:H`
> Score/severity: `7.8 / High`

Figure 1: Representative CTI mapping from a CVE description to structured labels and scores.

A key observation behind this work is that many CTI tasks are directly verifiable. Unlike open-ended preference tasks, CTI outputs often have canonical targets: an ATT&CK technique ID, a CWE label, a CVSS vector, a mitigation set, or a structured extraction schema. This makes CTI well-suited to reinforcement learning with verifiable rewards (RLVR), where deterministic programmatic verifiers score model outputs without requiring a learned reward model or human preference labels. RLVR has recently improved reasoning and structured generation in LLMs (Shao et al., 2024; DeepSeek-AI et al., 2025; Wen et al., 2025), while avoiding the cost and subjectivity of RLHF-style preference collection (Ouyang et al., 2022). However, standard on-policy RLVR is limited by empirical support: with a small rollout budget, hard prompts may produce no verified-correct completions, yielding little useful learning signal in that iteration (Wu et al., 2025). We find this sparse-reward regime common in CTI, especially for long-tail identifiers and strict structured outputs.

We introduce **Minerva**, a unified CTI training suite and RLVR pipeline for specializing open-weight LLMs to verifier-checkable CTI workflows. Minerva-CTI contains 16 training tasks spanning three broad families: vulnerability-centric mapping, such as CVE → CWE/CVSS/ATT&CK; detection-centric mapping, such as Sigma, Microsoft Sentinel, and Splunk rules → ATT&CK; and procedure-oriented mapping, such as scenarios or behaviors → techniques, tactics, mitigations, or threat actors. All tasks are normalized to canonical target spaces and paired with deterministic verifiers.

To train reliably under sparse verifier feedback, we propose **MinervaRL**. MinervaRL augments the standard GRPO-based RLVR loop with hardness-gated answer-conditioned rationale generation. When the current policy fails to produce a fully verified rollout for a prompt, MinervaRL temporarily reveals the gold label during training to elicit a short rationale trace, filters the generated candidates for correctness and quality, and distills accepted traces back onto the original answer-free prompt. At evaluation time, MinervaRL uses the same answer-free prompts as GRPO and receives no label hints.

Our contributions are:

- We curate **Minerva-CTI**, a unified 16-task CTI training suite with deterministic, verifier-checkable targets spanning vulnerability, detection, and procedure-oriented CTI workflows.

- We propose **MinervaRL**, an RLVR extension that mitigates sparse verifier feedback by leveraging hardness-gated, answer-conditioned rationale generation and periodic distillation onto the original answer-free prompts.

- We evaluate across 12 CTI benchmarks and four open-weight backbones. MinervaRL improves the mean CTI score by 15.8 percentage points over matched base checkpoints and by 4.3 points over GRPO, while outperforming controlled SFT, rejection-finetuning, and off-policy RLVR baselines on average.

## 2 Related Work

**LLMs for cyber threat intelligence.** LLMs have increasingly been applied to CTI workflows that require extracting, normalizing, and grounding security-relevant information from unstructured reports. A central task is mapping tactics, techniques, and procedures (TTPs) from natural-language threat reports to MITRE ATT&CK. TRAM demonstrates an applied LLM pipeline for automated technique identification, motivated by

the cost and brittleness of manual ATT&CK mapping (Center for Threat-Informed Defense, 2023). A recent systematization of automated TTP extraction methods, including generative LLMs, highlights persistent comparability challenges arising from heterogeneous ontologies, datasets, and evaluation protocols (Büchel et al., 2025). Expert-annotated resources such as AnnoCTR provide more controlled supervision and evaluation for ATT&CK-labeled CTI text (Lange et al., 2024). Beyond report-level extraction, LLMs have been used to bootstrap structured CTI artifacts such as knowledge graphs (Hu et al., 2024), and hybrid knowledge-graph/LLM systems have been proposed for producing actionable intelligence from heterogeneous evidence (Fieblinger et al., 2024). Other work maps vulnerability descriptions into standardized taxonomies, where prompting alone remains unreliable but instruction templating and fine-tuning can improve grounding (Liu et al., 2023; Zhang et al., 2024). Instruction-tuned security models such as CyberPal and CyberPal 2.0 further improve cybersecurity-oriented behavior, while also illustrating the limitations of supervised fine-tuning for robust CTI reasoning and structured output generation (Levi et al., 2024; 2025). Complementary efforts study data-efficient ATT&CK technique identification, including active learning (Rahman et al., 2024), and LLM pipelines for mapping detection artifacts, such as Sigma rules and SIEM analytics, to ATT&CK via prompt chaining and retrieval (Wudali et al., 2025).

**CTI and cybersecurity benchmarks.** A growing set of benchmarks evaluates LLMs on CTI and cybersecurity workflows beyond general NLP tasks. CTIBench and AthenaBench target CTI-specific capabilities such as CVE→CWE mapping, CVSS prediction, ATT&CK technique extraction, mitigation recommendation, and threat-actor attribution (Alam et al., 2024; 2025). SEvenLLM introduces a bilingual cybersecurity instruction corpus and benchmark with incident-analysis and response-oriented CTI tasks (Ji et al., 2024). Broader cybersecurity evaluations include SECURE, which measures LLM performance across multiple cybersecurity tasks (Bhusal et al., 2024); CyberMetric, which evaluates broad cybersecurity knowledge through 10,000 questions (Tihanyi et al., 2024); and CyberBench, which aggregates datasets for cybersecurity-language understanding tasks such as entity recognition, summarization, and classification (Liu et al., 2024). These benchmarks expose recurring failure modes, including hallucination, mis-grounding, brittle identifier mapping, and schema errors.

**Reinforcement learning with verifiable rewards.** Reinforcement learning with verifiable rewards (RLVR) has become a prominent post-training approach for tasks where correctness can be checked programmatically, especially mathematical reasoning and code generation (Shao et al., 2024; DeepSeek-AI et al., 2025). Instead of relying on learned preference models, RLVR uses deterministic verifiers to assign rewards, making it attractive for domains with canonical answers and structured outputs. Recent work shows that RLVR can improve reasoning behavior, reliability, and calibration relative to supervised baselines (Wen et al., 2025), while other studies examine whether RLVR elicits new capabilities or primarily amplifies behaviors already present in the base model (Yue et al., 2025; Cheng et al., 2025; Wu et al., 2025). Related analyses further suggest that supervised fine-tuning can overfit or memorize solution traces, whereas reinforcement learning may encourage more generalizable behavior under verifiable feedback (Chu et al., 2025). Follow-up work studies how RLVR gains depend on task difficulty, rollout budget, and exploration strategy (Yang et al., 2025), and shows that verifiable-reward formulations can transfer beyond math and code to other structured domains (Lu et al., 2025; Su et al., 2025). Our work brings this paradigm to CTI, where many targets are canonical identifiers, label sets, or structured strings, and addresses the sparse-reward regime through hardness-gated answer-conditioned rationale generation and original-prompt distillation.

## 3 Minerva-CTI Dataset

We introduce **Minerva-CTI**, a unified CTI training suite curated from standards, knowledge bases, and community-maintained security resources, including MITRE ATT&CK (MITRE Corporation, 2026d;a), MITRE CAPEC (MITRE Corporation, 2026b), the National Vulnerability Database (NVD) (Byers et al., 2022; MITRE Corporation, 2026c), Mappings Explorer (Center for Threat-Informed Defense, 2026), and detection or emulation corpora such as Sigma, Atomic Red Team, Microsoft Sentinel, and Splunk Security Content (SigmaHQ, 2026; Red Canary, 2026; Microsoft, 2026; Splunk, 2026). From these sources, we define 16 verifier-checkable tasks spanning vulnerability mapping, detection-rule mapping, and procedure-oriented CTI reasoning. Tasks include predicting ATT&CK techniques, tactics, and mitigations; mapping CVEs

to CWE labels and CVSS v3.1 vectors; attributing threat actors based on observed behaviors; and linking CAPEC examples to attack patterns or weaknesses. The dataset contains 32,000 training instances and 1,200 validation instances; Appendix A provides full task definitions, sources, and split statistics.

Each instance consists of an analyst-style prompt $x$, a canonical target $y^\star$, and task metadata used by the verifier. Targets include single identifiers, unordered identifier sets, and structured strings such as CVSS v3.1 vectors. During construction, we normalize labels against fixed taxonomy snapshots, map aliases to canonical forms when applicable, and deduplicate set-valued targets before scoring. This yields stable verifier targets despite heterogeneous surface forms across CTI sources. Minerva-CTI is intentionally heterogeneous: examples are allocated per task so that low-resource mappings remain represented alongside larger CVE- and scenario-derived tasks. The held-out Minerva validation split is used for checkpoint selection and diagnostics, while the 12-task suite in Section 5.1 evaluates both training-aligned performance and transfer to benchmarks not used as Minerva-CTI training objectives.

## 4 Methodology

### 4.1 Reinforcement Learning with Verifiable Rewards

We train CTI models using reinforcement learning with verifiable rewards (RLVR). Let $x \in \mathcal{X}$ denote a prompt, $y \in \mathcal{Y}$ a model-generated output sequence, and $y^\star \in \mathcal{A}$ the canonical target answer for $x$. Each training task is paired with a deterministic programmatic verifier $R_{\mathrm{MINERVA}} : \mathcal{X} \times \mathcal{Y} \times \mathcal{A} \to [0,1]$, which extracts the final answer from $y$, normalizes it, and scores it against $y^\star$. Rewards are instantiated as exact identifier matching for single-label outputs, structured partial credit for hierarchical labels such as ATT&CK techniques and sub-techniques, or set-based overlap for multi-label targets such as tactics, mitigations, and CWE sets. This formulation is well suited to CTI because many analyst-facing outputs are structured, canonicalized, and automatically auditable. It therefore enables scalable optimization without a learned reward model or human preference labels. Appendix B describes answer extraction, normalization, and task-specific verification. Unless otherwise stated, we do not add a separate format reward.

We optimize the policy $\pi_\theta$, parameterized by $\theta$, with Group Relative Policy Optimization (GRPO) (Shao et al., 2024; DeepSeek-AI et al., 2025), a PPO-style objective (Schulman et al., 2017) that avoids training a value critic by estimating advantages from multiple completions sampled for the same prompt. For each training pair $(x, y^\star)$, we sample a group of $N$ completions $\{y_j\}_{j=1}^N$ from the behavior policy $\pi_{\theta_{\mathrm{old}}}(\cdot \mid x)$, score them with the verifier to obtain rewards $r_j = R_{\mathrm{MINERVA}}(x, y_j, y^\star)$, and compute group-normalized advantages

$$\hat{A}_j = \frac{r_j - \mathrm{mean}(\{r_h\}_{h=1}^N)}{\mathrm{std}(\{r_h\}_{h=1}^N) + \epsilon}. \tag{1}$$

Here $\epsilon > 0$ is a small numerical stabilizer. The policy is then updated with the clipped GRPO surrogate

$$\mathcal{L}_{\mathrm{GRPO}}(\theta) = -\mathbb{E}\left[\frac{1}{N}\sum_{j=1}^N \min\left(\rho_j(\theta)\hat{A}_j, \mathrm{clip}(\rho_j(\theta), 1-\varepsilon, 1+\varepsilon)\,\hat{A}_j\right)\right], \quad \rho_j(\theta) = \frac{\pi_\theta(y_j \mid x)}{\pi_{\theta_{\mathrm{old}}}(y_j \mid x)}. \tag{2}$$

Here $\rho_j(\theta)$ is the likelihood ratio between the updated policy and the behavior policy for completion $y_j$, and $\varepsilon$ is the clipping radius. The expectation is over sampled training pairs and their completion groups. This critic-free objective uses only verifier scores and within-prompt relative comparisons, making it a natural optimizer for structured CTI tasks with automatically checkable outputs.

### 4.2 MinervaRL

Minerva-CTI tasks are verifier-checkable, but they are not uniformly easy for on-policy RLVR. Many tasks require selecting an exact identifier, or a small set of identifiers, from a large, finite, and long-tailed label space. For example, our January 2026 snapshots contain 944 CWE identifiers and 44 MITRE Enterprise mitigation identifiers, with many labels appearing rarely in public pretraining data compared to more common schemas, such as ATT&CK techniques. Under a small rollout budget, the policy may therefore fail to sample any fully

correct completion for hard prompts. In such cases, all completions in the group receive either no reward or only partial reward, and the resulting GRPO update provides little signal about the correct structured answer. This reward-sparsity regime motivates an auxiliary mechanism that can seed verified trajectories for hard prompts while preserving the original answer-free inference setting.

MinervaRL augments the standard GRPO-based RLVR loop with hardness-gated answer-conditioned self-training. The central idea is simple: when the current policy cannot solve a prompt within the available rollout budget, we temporarily reveal the ground-truth label during training to elicit a short rationale that justifies it. We call the resulting trace an *answer-conditioned rationale* (ACR). Importantly, the label-revealing prompt is used only to generate candidate training traces. Before distillation, each candidate is checked by the task verifier and filtered for leakage and rationale quality. Accepted traces are then distilled back into the actor using the original task prompt, without any label hint. Thus, MinervaRL uses labels to construct additional verified supervision during training, but all reported evaluations use the same answer-free prompts as the GRPO baseline.

The procedure runs on two coupled timescales. At every training step, the actor performs the ordinary on-policy GRPO update on verifier-scored rollouts from the original prompts, so the primary RLVR objective is unchanged. In parallel, MinervaRL identifies hard prompts whose rollout group contains no fully verified completion, i.e., prompts with maximum verifier reward below 1. These prompts are added to an ACR buffer $\mathcal{P}$ together with a label-conditioned generation prompt. Every $I$ steps, an exponential moving average (EMA) teacher samples ACR candidates for the buffered prompts. Candidates that are verifier-correct and pass the filtering pipeline are stored in a distillation buffer $\mathcal{Q}$ as pairs of the original prompt and the accepted trace. The actor then receives a lightweight supervised update on samples from $\mathcal{Q}$, after which the buffers are reset. In this way, MinervaRL preserves on-policy RLVR as the main training signal while periodically converting otherwise signal-poor hard prompts into verified original-prompt training examples.

**Notation.** Let $\mathcal{D} = \{(x_i, y_i^\star)\}_{i=1}^n$ be the Minerva-CTI training set, where $x_i \in \mathcal{X}$ is the original answer-free prompt and $y_i^\star \in \mathcal{A}$ is the ground-truth structured target. The actor $\pi_\theta$ and EMA teacher $\pi_\phi$ both generate output sequences $y \in \mathcal{Y}$, including ordinary RLVR responses and ACR traces. The buffers $\mathcal{P}$ and $\mathcal{Q}$ store ACR-generation prompts and accepted original-prompt distillation pairs, respectively. We write $x_i \oplus b(y_i^\star, d_i)$ for concatenating $x_i$ with a label-conditioned ACR suffix $b(\cdot)$, where $d_i$ denotes optional canonical reference details, and $\mathrm{task}(x_i)$ for the task type of prompt $x_i$.

**Step 1 (Algorithm 1): RLVR rollouts and GRPO update.** At step $t$, we sample a batch $\mathcal{B}_t \subset \mathcal{D}$. For each $(x_i, y_i^\star) \in \mathcal{B}_t$, the actor samples $N = 8$ completions from the original prompt,

$$y_{i,j}^{\mathrm{rlvr}} \sim \pi_\theta(\cdot \mid x_i), \qquad j = 1, \dots, N, \tag{3}$$

which are scored by the verifier:

$$r_{i,j}^{\mathrm{base}} = R_{\mathrm{MINERVA}}\left(x_i, y_{i,j}^{\mathrm{rlvr}}, y_i^\star\right). \tag{4}$$

The actor is updated with GRPO using only these on-policy, original-prompt trajectories, matching the standard GRPO baseline.

**Step 2 (Algorithm 1): Hard-prompt buffering.** For each prompt, compute the best rollout reward

$$m_i = \max_{j \in \{1, \dots, N\}} r_{i,j}^{\mathrm{base}}. \tag{5}$$

We call $x_i$ hard at step $t$ if $m_i < 1$, i.e., none of the sampled completions is fully verified. For each hard prompt, excluding CVSS, we construct an answer-conditioned rationale prompt

$$x_i^{\mathrm{acr}} = x_i \oplus b(y_i^\star, d_i), \tag{6}$$

where $b(\cdot)$ reveals the gold target and asks for a short rationale, and $d_i$ optionally contains truncated canonical reference details. We add $(x_i, x_i^{\mathrm{acr}}, y_i^\star)$ to the ACR buffer $\mathcal{P}$. This gate is dynamic: a prompt stops entering $\mathcal{P}$ once the actor begins producing fully verified rollouts. We exclude CVSS because its verifier already provides dense graded feedback through CVSS base-score distance, unlike long-tail identifier tasks where incorrect IDs often yield no useful signal.

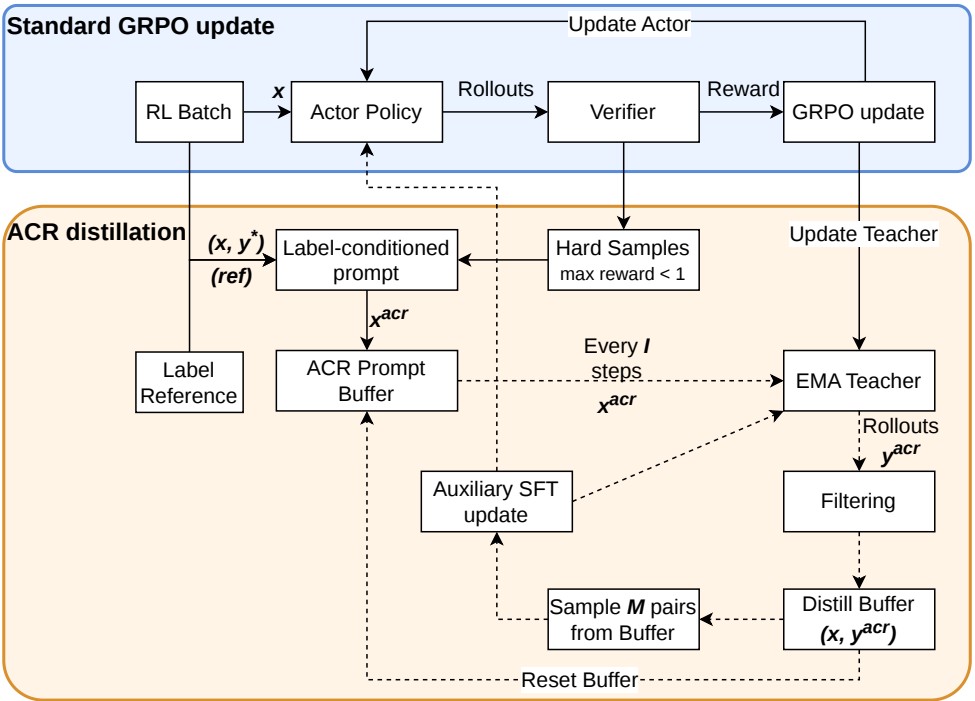

Figure 2: MinervaRL overview separating the standard GRPO loop from the auxiliary ACR path. Solid arrows show the synchronous training flow, while dashed arrows show the periodic ACR distillation pass.

**Step 3 (Algorithm 1): ACR generation and filtering.** Every $I = 10$ steps, the EMA teacher samples $K = 4$ ACR candidates for each buffered prompt:

$$y_{i,k}^{\mathrm{acr}} \sim \pi_\phi(\cdot \mid x_i^{\mathrm{acr}}), \qquad k = 1, \dots, K, \tag{7}$$

using temperature 0.7 and nucleus sampling with $p = 0.9$. We retain only candidates that are verifier-correct and pass the filtering pipeline in Appendix C, which removes leakage, degeneration, insufficiently grounded rationales, and low-quality traces. The eligible set is

$$\mathcal{E}_i = \left\{ k : R_{\mathrm{MINERVA}}\big(x_i, y_{i,k}^{\mathrm{acr}}, y_i^\star\big) = 1 \;\wedge\; \mathrm{Filter}\big(y_{i,k}^{\mathrm{acr}}\big) = 1 \;\wedge\; q_{i,k} \geq \tau_q \right\}, \tag{8}$$

where $q_{i,k}$ is the TextCNN GOOD probability and $\tau_q = 0.5$. If $\mathcal{E}_i \neq \varnothing$, we select

$$k^\star = \arg\max_{k \in \mathcal{E}_i} q_{i,k} \tag{9}$$

and enqueue the original-prompt distillation pair $(x_i, y_{i,k^\star}^{\mathrm{acr}})$ into $\mathcal{Q}$. Keeping at most one trace per prompt per interval prevents repeated generations from dominating the supervised update.

**Step 4 (Algorithm 1): Original-prompt distillation.** At each distillation interval, we sample up to $M = 256$ pairs $\mathcal{S} \subseteq \mathcal{Q}$ and apply one supervised update:

$$\mathcal{L}_{\mathrm{sft}}(\theta) = -\mathbb{E}_{(x,y) \sim \mathcal{S}} \left[ \log \pi_\theta(y \mid x) \right]. \tag{10}$$

The update uses learning rate $\gamma \cdot \mathrm{lr}_{\mathrm{rlvr}}$ with $\gamma = 0.05$, after which $\mathcal{P}$ and $\mathcal{Q}$ are reset. Because distillation is performed on the original prompt $x_i$, not on $x_i^{\mathrm{acr}}$, the actor learns from verified traces without relying on label hints at inference time.

---

**Algorithm 1** MinervaRL

---

**Require:** $\mathcal{D} = \{(x_i, y_i^\star)\}_{i=1}^n$; actor $\pi_\theta$; verifier $R_{\text{MINERVA}}$; RLVR rollouts $N$; ACR rollouts $K$; interval $I$; EMA decay $\alpha$;
distill cap $M$; threshold $\tau_q$; SFT scale $\gamma$

**Ensure:** Trained actor $\pi_\theta$

1: **Initialize:** $\phi \leftarrow \theta$; $\mathcal{P}, \mathcal{Q} \leftarrow \emptyset$
2: **for** training step $t = 1, 2, \ldots$ **do**
3:     **Step 1: RLVR rollouts and GRPO update**
4:     $\mathcal{B}_t \leftarrow \text{SAMPLEBATCH}(\mathcal{D})$
5:     **for all** $(x_i, y_i^\star) \in \mathcal{B}_t$ **do**
6:         $Y_i^{\text{rlvr}} \leftarrow \{y_{i,j}^{\text{rlvr}} \sim \pi_\theta(\cdot \mid x_i)\}_{j=1}^N$
7:         $\mathbf{r}_i^{\text{base}} \leftarrow \{r_{i,j}^{\text{base}} = R_{\text{MINERVA}}(x_i, y_{i,j}^{\text{rlvr}}, y_i^\star)\}_{j=1}^N$
8:         $m_i \leftarrow \max_j r_{i,j}^{\text{base}}$
9:     $\theta \leftarrow \text{GRPO}\big(\theta, \mathcal{B}_t, \{Y_i^{\text{rlvr}}, \mathbf{r}_i^{\text{base}}\}_i\big)$;    $\phi \leftarrow \alpha\phi + (1-\alpha)\theta$
10:    **Step 2: Buffer hard prompts for ACR**
11:    $\mathcal{P} \leftarrow \mathcal{P} \cup \{(x_i, x_i^{\text{acr}}, y_i^\star) : (x_i, y_i^\star) \in \mathcal{B}_t,\ x_i^{\text{acr}} = x_i \oplus b(y_i^\star, d_i), m_i < 1,\ \text{task}(x_i) \neq \text{CVSS}\}$
12:    **if** $t \bmod I = 0$ **then**
13:        **Step 3: ACR generation and filtering**
14:        **for all** $(x_i, x_i^{\text{acr}}, y_i^\star) \in \mathcal{P}$ **do**
15:           $Y_i^{\text{acr}} \leftarrow \{y_{i,k}^{\text{acr}} \sim \pi_\phi(\cdot \mid x_i^{\text{acr}})\}_{k=1}^K$
16:           $\mathbf{q}_i \leftarrow \{q_{i,k} = \text{TEXTCNNGOOD}(y_{i,k}^{\text{acr}})\}_{k=1}^K$
17:           $\mathcal{E}_i \leftarrow \{k : R_{\text{MINERVA}}(x_i, y_{i,k}^{\text{acr}}, y_i^\star) = 1 \wedge \text{Filter}(y_{i,k}^{\text{acr}}) = 1 \wedge q_{i,k} \geq \tau_q\}$
18:           **if** $\mathcal{E}_i \neq \emptyset$ **then**
19:              $k^\star \leftarrow \arg\max_{k \in \mathcal{E}_i} q_{i,k}$;    $\mathcal{Q} \leftarrow \mathcal{Q} \cup \{(x_i, y_{i,k^\star}^{\text{acr}})\}$
20:    **Step 4: Distill accepted traces on original prompts**
21:    $\mathcal{S} \leftarrow \text{SAMPLE}(\mathcal{Q}, M)$;    $\theta \leftarrow \text{SFT}(\theta, \mathcal{S}, \gamma \, \text{lr}_{\text{rlvr}})$;    $\mathcal{P}, \mathcal{Q} \leftarrow \emptyset$

---

## 4.3 Expanding Empirical Support with MinervaRL

Let $a^\star$ denote the correct structured answer for prompt $x$, and let

$$p_\theta(a^\star \mid x) := \Pr_{y \sim \pi_\theta(\cdot \mid x)}[g(y) = a^\star],$$

where $g(\cdot)$ is the task-specific answer extractor. With $k$ independent rollouts, the probability that the rollout group contains no verifier-correct completion is

$$\Pr[\text{no success in } k \text{ rollouts}] = (1 - p_\theta(a^\star \mid x))^k. \tag{11}$$

For a failure tolerance $\zeta \in (0, 1)$, define the detectability threshold

$$\varepsilon_{k,\zeta} := 1 - \zeta^{1/k} \approx \frac{-\log \zeta}{k}. \tag{12}$$

If $p_\theta(a^\star \mid x) < \varepsilon_{k,\zeta}$, then the correct answer is not reliably observed under the available rollout budget, so the prompt may repeatedly produce all-zero rollout groups and provide little direct positive signal to GRPO.

MinervaRL addresses this finite-sampling barrier by temporarily using the gold label during training to seed a verified trajectory for hard prompts. For a hard prompt, the answer-conditioned prompt $x^{\text{acr}} = x \oplus b(y^\star, d)$ makes it more likely that the EMA teacher will generate a verifier-correct trace. After verifier and quality filtering, the accepted trace is distilled back onto the original prompt $x$, without any label hint. This supervised step aims to increase $p_\theta(a^\star \mid x)$ under the original prompt so that future RLVR rollouts can observe verifier-correct completions under the same budget $k$. Thus, MinervaRL does not change the inference-time task or require label hints at deployment; it expands the set of prompts for which verifier feedback is empirically observable during training. This analysis isolates a finite-sampling support effect: ACR distillation can move hard prompts into the empirical support of a limited rollout budget. A conditional formalization of this support-expansion view, including assumptions, theorem statements, and proof sketches, is provided in Appendix J.

# 5 Experiments

## 5.1 Evaluation Benchmarks

We evaluate on the 12 CTI evaluation tasks in Table 1, covering multiple-choice cybersecurity and CTI knowledge, structured taxonomy prediction, SOC-style reasoning, and information extraction. These tasks include **CKT** (Alam et al., 2025), a five-option CTI knowledge QA benchmark; **CyberMetric** (Tihanyi et al., 2024), a four-option cybersecurity knowledge QA benchmark; **SOCEval** (Deason et al., 2025), multi-select SOC-style reasoning over threat-intelligence reports; **RCM** (Alam et al., 2025), CVE-to-CWE root-cause mapping; **VSP** (Alam et al., 2025), CVSS v3.1 base-vector prediction; **ATE** (Alam et al., 2025), ATT&CK technique identification from attack scenarios; **RMS** (Alam et al., 2025), ATT&CK mitigation recommendation; **ElasticRule** (Elastic, 2026), Elastic detection-rule to ATT&CK technique mapping; **APTNER** (Wang et al., 2022), APT-focused named-entity recognition; **LANCE** (Froudakis et al., 2025), IoC identification over IP, URL, domain, and hash candidates; **AnnoCTR** (Lange et al., 2024), STIX-style entity and relation extraction; and **AZERG** (Lekssays et al., 2025), STIX-style entity and relation extraction from CTI text. Full task definitions, output formats, and sample counts are provided in Appendix E.

**Contamination and overlap considerations.**   Because CTI benchmarks often draw on shared public standards and threat intelligence resources, fully eliminating train-test overlap is difficult. We therefore design Minerva-CTI to minimize direct leakage where possible. We exclude multiple-choice formats during training to reduce overlap with CKT and CyberMetric. SOCEval is derived from threat-intelligence reports that are not used as training targets. For CVE-based tasks, Minerva-CTI uses pre-2025 CVEs, while AthenaBench RCM and VSP evaluations emphasize 2025-era entries. For ATT&CK scenario tasks, Minerva-CTI uses ATT&CK-derived training scenarios, whereas ATE and RMS evaluation use independently curated model-generated scenarios conditioned on technique descriptions. Finally, ElasticRule, APTNER, LANCE, AnnoCTR, and AZERG are used only for evaluation and are not included as supervised training objectives.

## 5.2 Experimental Settings

**RLVR and MinervaRL training.**   All RLVR models use GRPO with batch size 128, $N = 8$ on-policy rollouts per prompt, 2048-token prompt truncation, 1024-token response truncation, actor learning rate $1 \times 10^{-6}$, and 500 training steps. Checkpoints for GRPO, MinervaRL, and the controlled trained baselines are selected using the same validation criterion: average performance on the held-out Minerva validation split (Minerva-Dev) and AthenaBench-Mini (Alam et al., 2025). MinervaRL uses the same GRPO loop, but for hard prompts with no fully verified rollout, an EMA teacher ($\alpha = 0.995$) samples $K = 4$ ACR traces at temperature 0.7 and nucleus sampling $p = 0.9$ using a 4096-token ACR context. Every $I = 10$ steps, up to $M = 256$ accepted traces are distilled onto the original answer-free prompts with learning rate $\gamma \cdot \mathrm{lr}_{\mathrm{rlvr}}$, where $\gamma = 0.05$. Full implementation details are in Appendix K.

**LLM backbones.**   We evaluate four open-weight backbones: **Llama-3.1-8B-Instruct** (Meta, 2024a), **Llama-3.2-3B-Instruct** (Meta, 2024b), **Qwen3-4B-Base** (Qwen Team, 2025a), and **Qwen3-8B-Base** (Qwen Team, 2025b). For each backbone, we report the unadapted base model, a GRPO-trained RLVR model, and a MinervaRL model trained with the same GRPO setup plus hardness-gated ACR generation and original-prompt distillation.

**Baselines.**   We compare against three controlled baselines using the same Minerva-CTI training split and verifiers: **STaR-CTI**, which iteratively bootstraps verifier-correct traces and retrains with SFT (Zelikman et al., 2022); **DART-CTI**, a rejection-finetuning baseline with a fixed verifier-filtered trace corpus (Tong et al., 2024); and **LUFFY-CTI**, an off-policy RLVR baseline using one accepted guidance trace per prompt (Yan et al., 2025). We also include two external Llama-3.1-8B security-SFT reference models, **Llama-Primus-Merged** and **Foundation-Sec-8B-Instruct** (Llama-Primus Team, 2025; Foundation AI, 2025). Baseline construction details are provided in Appendix H.

Table 1: Benchmark results across 12 CTI evaluation tasks. We compare MinervaRL with matched base models, GRPO, public security-SFT baselines, and STaR-CTI, DART-CTI, and LUFFY-CTI baselines. Underlined values indicate the best score within each backbone group.

| Model | CKT | CyberMetric | SOCEval | RCM | VSP | ATE | RMS | ElasticRule | APTNER | LANCE | AnnoCTR | AZERG | Avg. |
|---|---|---|---|---|---|---|---|---|---|---|---|---|---|
| Llama-3.1-8B-Instruct | 67.6 | 83.2 | 64.8 | 48.4 | 76.0 | 17.4 | 6.7 | 14.4 | 33.5 | 78.2 | 50.7 | 42.8 | 48.6 |
| Llama-Primus-Merged | 76.5 | 85.9 | 68.0 | 56.0 | 72.6 | 33.6 | 7.8 | 27.3 | 22.0 | 59.4 | 51.8 | 40.9 | 50.2 |
| Foundation-Sec-8B-Instruct | 77.1 | 81.3 | 67.9 | 61.0 | 65.8 | 39.2 | 15.4 | 33.6 | 34.0 | 59.5 | 43.5 | 44.4 | 51.9 |
| Llama-3.1-8B-STaR-CTI | 70.0 | 81.8 | 61.7 | 57.4 | 73.8 | 29.2 | 9.6 | 21.3 | 33.1 | 80.0 | 46.5 | 32.1 | 49.7 |
| Llama-3.1-8B-DART-CTI | 71.3 | 82.4 | 61.5 | 64.0 | 73.0 | 37.8 | 27.7 | 31.0 | 34.0 | 74.8 | 47.1 | 39.3 | 53.7 |
| Llama-3.1-8B-GRPO | 71.9 | 85.4 | 63.0 | 66.3 | 82.6 | 32.0 | 30.9 | 32.2 | 32.7 | 86.2 | 49.5 | 39.5 | 56.0 |
| Llama-3.1-8B-LUFFY-CTI | 73.9 | 83.7 | 63.3 | 63.8 | 79.6 | 40.6 | 34.3 | 33.8 | 35.4 | 82.4 | 49.7 | 43.1 | 57.0 |
| Llama-3.1-8B-MinervaRL | 73.9 | 84.2 | 64.7 | 68.8 | 87.6 | 48.4 | 42.1 | 40.5 | 34.1 | 84.6 | 50.3 | 43.7 | 60.2 |
| Llama-3.2-3B-Instruct | 71.6 | 77.0 | 55.7 | 15.2 | 2.8 | 1.0 | 0.4 | 1.2 | 25.4 | 77.8 | 35.9 | 32.8 | 33.1 |
| Llama-3.2-3B-STaR-CTI | 64.3 | 70.5 | 50.3 | 36.4 | 53.4 | 4.2 | 7.0 | 1.9 | 21.4 | 76.3 | 33.8 | 28.0 | 37.3 |
| Llama-3.2-3B-DART-CTI | 73.0 | 78.2 | 54.5 | 54.9 | 61.2 | 16.8 | 17.8 | 11.6 | 22.5 | 73.8 | 38.7 | 25.7 | 44.0 |
| Llama-3.2-3B-GRPO | 71.2 | 77.8 | 58.3 | 48.9 | 56.5 | 5.2 | 15.4 | 5.3 | 27.5 | 70.6 | 38.5 | 29.7 | 42.1 |
| Llama-3.2-3B-LUFFY-CTI | 70.6 | 78.5 | 55.4 | 43.8 | 71.6 | 19.4 | 24.2 | 20.4 | 15.0 | 68.0 | 46.0 | 32.1 | 45.4 |
| Llama-3.2-3B-MinervaRL | 71.0 | 78.1 | 54.6 | 57.2 | 77.1 | 21.8 | 29.3 | 20.1 | 16.7 | 82.2 | 37.9 | 33.7 | 48.3 |
| Qwen3-8B-Base | 71.7 | 69.3 | 63.8 | 52.6 | 69.0 | 13.4 | 6.5 | 9.0 | 31.3 | 45.1 | 9.8 | 9.8 | 37.6 |
| Qwen3-8B-STaR-CTI | 37.9 | 49.3 | 65.1 | 36.8 | 68.7 | 15.6 | 3.5 | 4.9 | 37.9 | 55.7 | 40.0 | 26.4 | 36.8 |
| Qwen3-8B-DART-CTI | 39.2 | 55.7 | 65.2 | 47.1 | 72.7 | 14.0 | 6.5 | 16.4 | 37.9 | 76.4 | 46.5 | 36.6 | 42.9 |
| Qwen3-8B-GRPO | 69.8 | 72.7 | 69.0 | 60.5 | 68.8 | 23.6 | 8.2 | 18.1 | 37.7 | 66.1 | 37.8 | 31.1 | 47.0 |
| Qwen3-8B-LUFFY-CTI | 78.6 | 88.3 | 64.8 | 65.3 | 73.2 | 29.4 | 25.8 | 32.4 | 34.3 | 74.5 | 45.1 | 26.8 | 53.2 |
| Qwen3-8B-MinervaRL | 77.8 | 88.2 | 67.5 | 64.8 | 79.4 | 32.0 | 20.1 | 22.0 | 37.4 | 74.9 | 43.0 | 33.1 | 53.4 |
| Qwen3-4B-Base | 45.6 | 50.1 | 56.1 | 45.6 | 76.5 | 4.2 | 6.5 | 4.6 | 1.2 | 18.0 | 16.6 | 6.5 | 27.6 |
| Qwen3-4B-STaR-CTI | 76.1 | 88.2 | 66.1 | 21.1 | 80.2 | 5.8 | 6.6 | 12.3 | 31.8 | 62.0 | 46.9 | 47.7 | 45.4 |
| Qwen3-4B-DART-CTI | 45.4 | 42.1 | 60.0 | 54.6 | 73.8 | 20.6 | 5.8 | 17.1 | 27.6 | 47.5 | 43.2 | 35.1 | 39.4 |
| Qwen3-4B-GRPO | 74.2 | 85.8 | 63.9 | 60.8 | 88.1 | 20.2 | 3.8 | 20.4 | 32.4 | 58.2 | 39.6 | 24.0 | 47.6 |
| Qwen3-4B-LUFFY-CTI | 54.1 | 56.5 | 60.9 | 59.9 | 72.6 | 17.6 | 10.3 | 23.4 | 29.2 | 60.4 | 46.7 | 36.2 | 44.0 |
| Qwen3-4B-MinervaRL | 70.0 | 80.0 | 64.0 | 59.9 | 80.0 | 25.4 | 5.1 | 27.5 | 28.0 | 56.7 | 50.4 | 29.1 | 48.0 |

**Evaluation metrics.** We use each benchmark's official or task-specific metric. Multiple-choice tasks use exact-match accuracy; taxonomy and mapping tasks use exact match on extracted ATT&CK or CWE labels; APTNER uses micro-F1 over JSON entities; RMS uses multi-label F1 over mitigation sets; and LANCE, AnnoCTR, and AZERG report averages over subtasks. For VSP, we follow the benchmark verifier and report the normalized CVSS score $1 - \mathrm{MAD}/7.7$, where MAD is the mean absolute deviation between predicted and gold CVSS v3.1 base scores.

**Evaluation protocol.** For the main single-sample results in Table 1, we use greedy decoding with temperature 0.0 and a maximum of 2048 new tokens. Each evaluation example is evaluated with its fixed task prompt, without model-specific prompt rewriting. The prompts specify the required output format, summarized in Table 15, such as a multiple-choice letter, CWE/ATT&CK/CVSS identifier, JSON object in `<json_object>` tags, or XML-like entity/relation tags. We score every system with the same task-specific parser, normalizer, and scoring script: the evaluator extracts the requested final answer or structured field using regex, JSON, or tag parsing as appropriate, canonicalizes identifiers where needed, and treats unparseable outputs as empty predictions. These rules are shared across all systems, including the external security-SFT baselines. The rollout-aware best-of-$k$ analysis in Section 7.2 uses a separate sampled decoding protocol.

**Uncertainty estimates.** To quantify evaluation uncertainty, we compute paired bootstrap 95% confidence intervals over evaluation examples. For each task, backbone, and baseline comparison, we resample task examples with replacement, recompute the original task metric for MinervaRL and the baseline on the same resampled examples, and report the percentile interval of the paired difference, MinervaRL minus the baseline. Dataset sizes for all evaluation tasks are listed in Table 15.

# 6 CTI Benchmark Results

## 6.1 Task-Level Results and Paired Comparisons

**Aggregate task scores.** Table 1 reports absolute scores across the 12 CTI evaluation tasks and four open-weight backbone families. Standard GRPO improves the average score over each matched base model,

Table 2: Per-task paired bootstrap 95% confidence intervals for MinervaRL minus GRPO, in percentage points. Positive intervals indicate tasks where MinervaRL is higher than GRPO under the paired bootstrap interval.

| Task | Llama-3.1-8B | Llama-3.2-3B | Qwen3-8B | Qwen3-4B |
|---|---|---|---|---|
| CKT | +2.0 [0.6, 3.5] | -0.2 [-1.6, 1.1] | +8.0 [6.5, 9.5] | -4.2 [-5.8, -2.6] |
| CyberMetric | -1.1 [-2.5, 0.3] | +0.3 [-1.1, 1.5] | +15.5 [13.5, 17.5] | -5.7 [-7.4, -4.0] |
| SOCEval | +1.7 [-0.4, 3.8] | -3.7 [-5.8, -1.4] | -1.4 [-2.7, -0.1] | +0.2 [-1.6, 2.1] |
| RCM | +2.4 [1.1, 3.8] | +8.3 [6.8, 9.8] | +4.3 [2.9, 5.8] | -0.9 [-2.1, 0.2] |
| VSP | +5.0 [4.3, 5.7] | +20.6 [18.9, 22.4] | +10.6 [9.7, 11.7] | -8.1 [-9.0, -7.3] |
| ATE | +16.4 [12.6, 20.4] | +16.6 [13.2, 20.2] | +8.4 [5.8, 11.4] | +5.2 [2.8, 7.6] |
| RMS | +11.2 [7.8, 14.4] | +13.9 [11.1, 16.9] | +12.0 [9.1, 14.8] | +1.3 [-0.7, 3.2] |
| ElasticRule | +8.3 [3.9, 12.7] | +14.8 [11.6, 18.3] | +3.9 [-0.2, 7.9] | +7.2 [3.7, 10.9] |
| APTNER | +1.4 [0.5, 2.4] | -10.8 [-12.3, -9.3] | -0.3 [-1.5, 1.0] | -4.4 [-5.9, -2.9] |
| LANCE | -1.5 [-5.7, 2.2] | +11.7 [2.7, 19.3] | +8.8 [0.8, 19.9] | -1.5 [-10.5, 9.3] |
| AnnoCTR | +0.7 [-1.6, 2.9] | -0.7 [-2.4, 1.1] | +5.2 [3.2, 8.1] | +10.7 [4.4, 12.9] |
| AZERG | +4.1 [2.3, 7.7] | +4.0 [1.5, 7.0] | +2.0 [0.1, 5.6] | +5.1 [0.3, 7.8] |

Table 3: Pairwise task-count summary for MinervaRL versus each baseline across 48 backbone-task comparisons. CI Pos. and CI Neg. are counts of comparisons whose paired 95% confidence interval is entirely above or below zero.

| Baseline | Point wins | Point losses | CI Pos. | CI Neg. | CI Overlap |
|---|---|---|---|---|---|
| Base | 42/48 | 6/48 | 35/48 | 2/48 | 11/48 |
| STaR | 38/48 | 10/48 | 35/48 | 6/48 | 7/48 |
| DART | 38/48 | 10/48 | 34/48 | 3/48 | 11/48 |
| GRPO | 34/48 | 14/48 | 28/48 | 7/48 | 13/48 |
| LUFFY | 32/48 | 16/48 | 20/48 | 6/48 | 22/48 |

raising Llama-3.1-8B from 48.6 to 56.0, Llama-3.2-3B from 33.1 to 42.1, Qwen3-8B from 37.6 to 47.0, and Qwen3-4B from 27.6 to 47.6. These gains are largest on structured verifier-aligned tasks such as CWE mapping, CVSS prediction, ATT&CK technique identification, mitigation recommendation, and rule-to-technique mapping. MinervaRL further raises the average to 60.2, 48.3, 53.4, and 48.0 for the four backbones, respectively, obtaining the highest average score within each backbone group.

**Comparison with controlled baselines.** Compared with STaR-CTI and DART-CTI, MinervaRL has a higher average score for all four backbone groups. LUFFY-CTI is the closest controlled baseline: MinervaRL is higher by +3.2 points on Llama-3.1-8B, +2.9 on Llama-3.2-3B, +0.2 on Qwen3-8B, and +4.0 on Qwen3-4B. On Llama-3.1-8B, MinervaRL also exceeds the two external security-SFT reference models, reaching 60.2 average compared with 50.2 for Llama-Primus-Merged and 51.9 for Foundation-Sec-8B-Instruct.

**Per-task paired uncertainty versus GRPO.** To complement the aggregate scores, Table 2 reports per-task paired bootstrap 95% confidence intervals over evaluation examples for MinervaRL minus GRPO. Across the 48 backbone-task comparisons, MinervaRL has positive point deltas on 34/48 comparisons; the paired interval is entirely above zero on 28/48, entirely below zero on 7/48, and overlaps zero on 13/48. The strongest CI-separated gains are concentrated on structured verifier-aligned tasks including RCM, VSP, ATE, RMS, and ElasticRule. The Qwen3-4B comparison is the least separated: its average gain over GRPO is small, and several task-level intervals are negative or overlap zero.

**Pairwise summary across baselines.** Table 3 summarizes paired interval signs for MinervaRL against each baseline across all 48 backbone-task comparisons. MinervaRL is higher than the base models on 42/48 point comparisons, with 35/48 CI-positive intervals. It also has broad CI-positive majorities against STaR-CTI and DART-CTI, with 35/48 and 34/48 CI-positive intervals, respectively. Against LUFFY-CTI, the closest controlled baseline, MinervaRL has positive point deltas on 32/48 comparisons; the paired 95% interval is entirely above zero on 20/48, entirely below zero on 6/48, and overlaps zero on 22/48.

Table 4: Task-family count summary for MinervaRL versus each baseline. Counts are over backbone-task comparisons within each family. Point W/L reports MinervaRL point wins/losses; CI +/-/0 reports CI-positive, CI-negative, and CI-overlap counts.

| Baseline | QA / selection (n=12) Point W/L | CI +/-/0 | Taxonomy mapping (n=16) Point W/L | CI +/-/0 | Vulnerability scoring (n=4) Point W/L | CI +/-/0 | Information extraction (n=16) Point W/L | CI +/-/0 |
|---|---|---|---|---|---|---|---|---|
| Base | 9/3 | 7/0/5 | 15/1 | 15/1/0 | 4/0 | 4/0/0 | 14/2 | 9/1/6 |
| STaR | 9/3 | 9/2/1 | 15/1 | 15/1/0 | 3/1 | 3/0/1 | 11/5 | 8/3/5 |
| DART | 10/2 | 9/1/2 | 15/1 | 15/0/1 | 4/0 | 4/0/0 | 9/7 | 6/2/8 |
| GRPO | 6/6 | 3/4/5 | 15/1 | 13/0/3 | 3/1 | 3/1/0 | 10/6 | 9/2/5 |
| LUFFY | 8/4 | 4/0/8 | 10/6 | 8/3/5 | 4/0 | 4/0/0 | 10/6 | 4/3/9 |

Table 5: Llama-3.1-8B component ablations on validation splits and the 12 CTI evaluation tasks. Validation columns report Minerva-Dev and AthenaBench-Mini scores. Evaluation columns report the 12-task macro average and paired bootstrap interval sign counts across the 12 tasks. Underlined values indicate the best score in each validation or 12-task average column.

| Ablation | Validation Minerva-Dev | AthenaBench-Mini | Avg. | 12 CTI evaluation tasks Avg. | CI Pos. | CI Neg. | CI Overlap |
|---|---|---|---|---|---|---|---|
| GRPO | 50.3 | 57.4 | 53.8 | 56.0 | 8 | 0 | 4 |
| GRPO 12 rollouts | 50.9 | 52.7 | 51.8 | 53.9 | 7 | 1 | 4 |
| Answer-only SFT | 58.6 | 38.7 | 48.7 | 52.6 | 7 | 3 | 2 |
| In-loop answer SFT | 58.3 | 57.8 | 58.0 | 57.8 | 6 | 4 | 2 |
| No EMA teacher | 63.1 | 62.2 | 62.6 | 57.2 | 8 | 0 | 4 |
| No TextCNN filter | 64.2 | 59.0 | 61.6 | 57.7 | 7 | 1 | 4 |
| No filtering | 63.0 | 61.4 | 62.2 | 57.8 | 7 | 0 | 5 |
| Full MinervaRL | 63.2 | 63.3 | 63.3 | 60.2 | – | – | – |

## 6.2 Task-Family Summary

To summarize the heterogeneous CTI evaluation tasks by output and scoring type, Table 4 groups the pairwise comparisons from Table 3 into four task families: QA/selection (CKT, CyberMetric, SOCEval), taxonomy mapping (RCM, ATE, RMS, ElasticRule), vulnerability scoring (VSP), and information extraction (APTNER, LANCE, AnnoCTR, AZERG). The denominators count task-by-backbone comparisons. By point wins, MinervaRL has a majority in every family-baseline block except QA/selection versus GRPO, where the comparison is tied at 6 wins and 6 losses. The same exception appears under CI-sign counts: CI-positive counts exceed CI-negative counts in every family-baseline block except QA/selection versus GRPO, where the signs are 3 positive, 4 negative, and 5 overlapping. The strongest interval-separated gains remain on taxonomy mapping and vulnerability scoring, while QA/selection and extraction tasks are more mixed.

## 6.3 Component Ablations

We report Llama-3.1-8B component ablations on the validation splits used during training, Minerva-Dev and AthenaBench-Mini, and also evaluate the resulting checkpoints on the 12 CTI evaluation tasks from Table 1. Table 5 reports validation scores, 12-task averages, and paired bootstrap sign counts for Full MinervaRL minus each control across the 12 task metrics. Table 27 reports the corresponding per-task scores.

The full method has the highest 12-task average among the component controls. In Table 5, GRPO is the no-ACR control, GRPO 12 rollouts controls for additional rollout sampling, and in-loop answer SFT matches the auxiliary distillation schedule while replacing ACR traces with direct final-answer targets. GRPO is 4.2 points lower, indicating that the gain is not explained by verifier RL alone. GRPO with 12 rollouts does not close the gap despite additional rollout compute, and in-loop answer SFT remains below full MinervaRL, suggesting that full ACR distillation is not reducible to direct final-answer SFT with matched auxiliary updates. Removing the EMA teacher or filtering components yields smaller but consistent drops in the 12-task average.

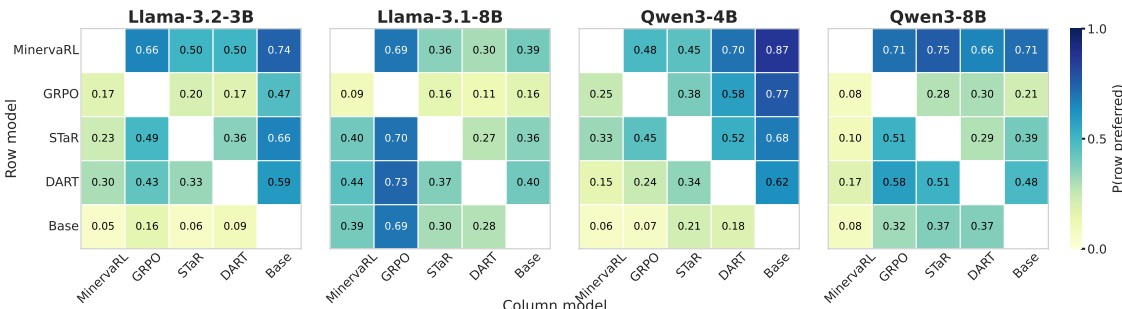

Figure 3: GPT-5.2 pairwise response-quality preferences for backbone-specific model groups. Each cell reports the probability that the row model is preferred over the column model, with ties retained in the denominator.

## 6.4 Response-Quality Preference Evaluation

Verifier-based scores measure structured CTI correctness, but analyst-facing usefulness also depends on readability, evidence use, and CTI concept precision. We therefore run a blind GPT-5.2 response-quality evaluation within each backbone family, comparing the base model, GRPO, MinervaRL, STaR-CTI, and DART-CTI. Responses are scored with a three-criterion rubric covering writing quality, prompt-evidence use, and CTI concept use; pairwise preferences are derived from total scores, with ties retained. Full prompts, rubrics, and validation details are in Appendix I.

**Preference results.** Figure 3 shows that MinervaRL is preferred over GRPO for all four backbones and is top-ranked for Llama-3.2-3B, Qwen3-4B, and Qwen3-8B. On Llama-3.1-8B, STaR-CTI and DART-CTI receive higher preference rates, suggesting that fixed-trace SFT and rejection-finetuning baselines can produce stronger prose for a strong instruction-tuned backbone. MinervaRL nevertheless remains preferred over the base and GRPO variants and achieves the highest average task score in Table 1, indicating that its verifier-score gains generally coincide with improved judged response quality relative to standard GRPO.

**Judge validation.** We validate GPT-5.2 on a 100-example subset over the five Llama-3.1-8B-family variants, using two human annotators and Claude Sonnet 4.6. GPT-5.2 agrees substantially with the human annotator average (QWK 0.6313, Spearman 0.7722), close to human–human agreement (QWK 0.6514, Spearman 0.6461), and also agrees strongly with Claude Sonnet 4.6 (QWK 0.7634, Spearman 0.8266). Full validation results are in Appendix I.

## 6.5 Training-Aligned vs. Not-in-Training Tasks

We split the 12 CTI evaluation tasks into tasks that directly match Minerva-CTI training objectives and tasks that are not directly optimized during training. The training-aligned group contains RCM, VSP, ATE, and RMS, corresponding to CVE-to-CWE mapping, CVSS vector prediction, ATT&CK technique prediction, and mitigation recommendation. The not-in-training group contains CKT, CyberMetric, SOCEval, ElasticRule, APTNER, LANCE, AnnoCTR, and AZERG. Table 6 reports the mean score for each group.

MinervaRL delivers the largest gains on training-aligned tasks, improving over the base model by +24.6, +41.5, +13.7, and +9.4 points on Llama-3.1-8B, Llama-3.2-3B, Qwen3-8B, and Qwen3-4B, respectively. It also improves over GRPO on three of the four backbones in this group. On not-in-training tasks, MinervaRL improves over GRPO for every backbone in the group average, with gains of +2.0, +1.9, +5.2, and +0.9 points. The task-count summary in Table 7 shows the same pattern at the task level: gains over the base model remain broad even outside the training objectives, while gains over GRPO are positive on a majority of not-in-training comparisons but more mixed than on training-aligned tasks. Overall, the gains are largest on verifier-aligned structured tasks; outside the training objectives, improvements are broader relative to base models and more mixed relative to GRPO.

Table 6: Average performance on training-aligned and not-in-training evaluation tasks.

| Model | Llama-3.1-8B | | Llama-3.2-3B | | Qwen3-8B | | Qwen3-4B | |
|---|---|---|---|---|---|---|---|---|
| | Aligned | Not-in-train | Aligned | Not-in-train | Aligned | Not-in-train | Aligned | Not-in-train |
| Base | 37.1 | 54.4 | 4.9 | 47.2 | 35.4 | 38.7 | 33.2 | 24.8 |
| GRPO | 53.0 | 57.5 | 31.5 | 47.4 | 40.3 | 50.3 | 43.2 | 49.8 |
| MinervaRL | 61.7 | 59.5 | 46.4 | 49.3 | 49.1 | 55.5 | 42.6 | 50.7 |

Table 7: Task-count summary for training-aligned and not-in-training tasks among the 12 CTI evaluation tasks. Counts are over backbone-task comparisons. Aligned tasks are RCM, VSP, ATE, and RMS; not-in-training tasks are the remaining eight tasks. CI Pos./Neg./Overlap indicate whether the paired bootstrap 95% confidence interval for MinervaRL minus the comparison model is entirely above zero, below zero, or overlaps zero.

| | Training-aligned | | | | | Not-in-training | | | | |
|---|---|---|---|---|---|---|---|---|---|---|
| Comparison | Point wins | Point losses | CI Pos. | CI Neg. | CI Overlap | Point wins | Point losses | CI Pos. | CI Neg. | CI Overlap |
| MinervaRL vs Base | 15/16 | 1/16 | 15/16 | 1/16 | 0/16 | 27/32 | 5/32 | 21/32 | 1/32 | 10/32 |
| MinervaRL vs GRPO | 14/16 | 2/16 | 13/16 | 1/16 | 2/16 | 20/32 | 12/32 | 15/32 | 6/32 | 11/32 |

# 7 Training Dynamics and Additional Evaluations

## 7.1 Reward Sparsity and Validation Dynamics

A central challenge in CTI RLVR is reward sparsity: for some prompts, none of the sampled rollouts receives any verifier reward, leaving GRPO with little direct learning signal for that example. We track this failure mode using the zero-solve fraction, defined as the fraction of prompts whose rollout group has maximum verifier reward equal to 0. As shown in Figure 4, MinervaRL reduces the zero-solve fraction relative to GRPO across all four backbone families. For Llama-3.1-8B, increasing the GRPO rollout budget from $N = 8$ to $N = 12$ also reduces zero-solve frequency, but does not close the gap to MinervaRL. This suggests that MinervaRL's gains are not merely due to additional sampling; answer-conditioned trace generation and original-prompt distillation help the policy reach verifier-detectable outputs under the same sparse-reward setting.

Figure 5 shows that the reduction in zero-solve prompts is accompanied by improved validation performance. Across all four backbones, MinervaRL reaches the best GRPO validation accuracy during training and then continues improving beyond it. This indicates that the support-seeding mechanism does more than change rollout diversity: it converts sparse-reward prompts into useful training signal that improves verifier-scored validation accuracy.

## 7.2 Rollout-Aware Best-of-$k$ Evaluation

We further evaluate GRPO and MinervaRL on the 12 CTI evaluation tasks under matched rollout budgets: for each example, we sample eight responses from each model using temperature 0.7, top-$p$ 0.95, and a 2048-token generation limit, then compute best-of-$k$ scores for $k \in \{1, 2, 4, 8\}$. For exact-match, MCQ, and taxonomy tasks, best-of-$k$ is the fraction of examples with at least one verified correct response. For graded, extraction, and multi-label tasks, we select the highest-scoring rollout for each example and then compute the task metric on those selected outputs. Table 8 summarizes paired comparisons across all backbone-task settings, Table 26 reports the full task scores, and Tables 22 to 25 report the per-task paired intervals.

MinervaRL improves over GRPO on 32/48, 36/48, 38/48, and 39/48 backbone-task comparisons for $k = 1, 2, 4, 8$, respectively. At $k = 8$, the paired bootstrap interval is entirely above zero for 31/48 comparisons, entirely below zero for 6/48, and overlaps zero for 11/48. As the rollout budget increases, MinervaRL's relative advantage over GRPO becomes slightly broader: point wins rise from 32/48 at $k = 1$ to 39/48 at $k = 8$, and CI-positive comparisons are highest at $k = 4$ and $k = 8$.

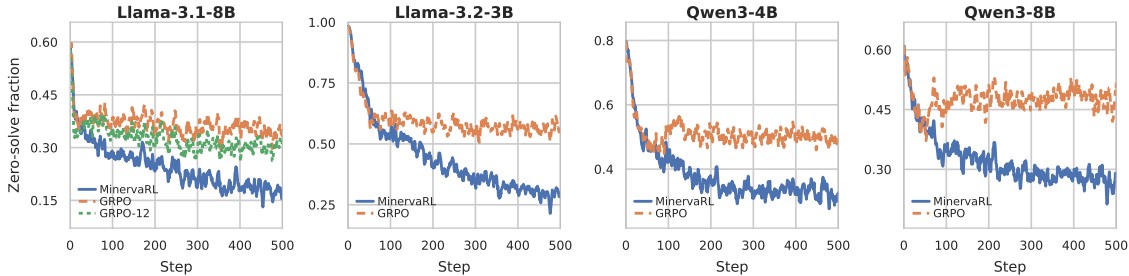

Figure 4: Fraction of prompts whose rollout group has maximum verifier reward $= 0$. Each panel compares GRPO and MinervaRL; Llama-3.1-8B also includes GRPO-12 ($N = 12$). Curves use a 5-step rolling mean.

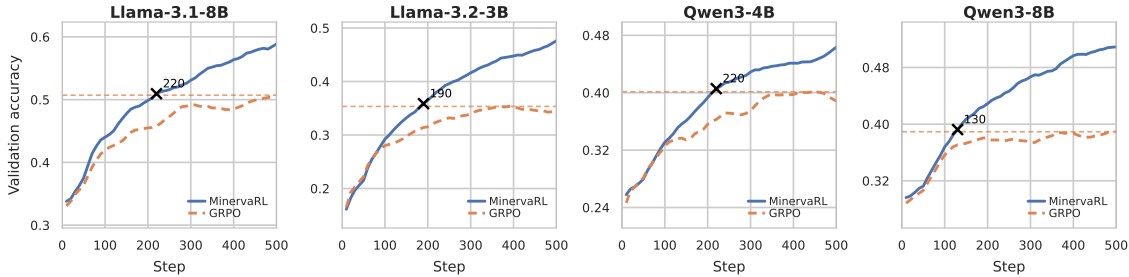

Figure 5: Validation accuracy over training. Curves are 5-step rolling means; dashed lines mark the best GRPO accuracy, and crosses mark the first MinervaRL step that reaches it.

Table 8: Rollout-aware task-count summary for MinervaRL versus GRPO across 48 backbone-task comparisons at each best-of-$k$ budget. CI Pos. and CI Neg. are counts of comparisons whose paired 95% confidence interval is entirely above or below zero.

| $k$ | Point wins | Point losses | CI Pos. | CI Neg. | CI Overlap |
|---|---|---|---|---|---|
| 1 | 32/48 | 16/48 | 27/48 | 10/48 | 11/48 |
| 2 | 36/48 | 12/48 | 26/48 | 6/48 | 16/48 |
| 4 | 38/48 | 10/48 | 31/48 | 4/48 | 13/48 |
| 8 | 39/48 | 9/48 | 31/48 | 6/48 | 11/48 |

### 7.3 General Capability and Structured-Output Transfer

We include two auxiliary evaluations: general/cyber benchmarks for Llama-3.1-8B variants and a preliminary text-to-SQL transfer experiment. Table 9(a) shows that MinervaRL largely preserves MMLU-Pro (Wang et al., 2024) and IFEval (Zhou et al., 2023) performance, improves CanaryExploit (Bhatt et al., 2024) over the base and GRPO variants, and does not increase WMDP-Cyber (Li et al., 2024) relative to GRPO, where lower scores indicate lower measured cyber-risk.

Table 9(b) evaluates transfer to text-to-SQL, another structured-output setting with executable or verifier-style rewards. Starting from Qwen2.5-Coder-3B, we train GRPO and MinervaRL using the SQL-R1 (Ma et al., 2025) training/validation setup and compare against the base model and SQL-R1-3B. MinervaRL obtains the best average score (60.8) and leads on four of seven datasets, providing preliminary evidence that the MinervaRL training recipe can transfer to another structured-output setting with verifier-style rewards.

### 7.4 Hyperparameter Sensitivity

Table 10 shows sensitivity to the distillation learning-rate scale $\gamma$ used in the supervised update, with learning rate $\gamma \cdot \mathrm{lr_{rlvr}}$. Performance is best around $\gamma = 0.05$: smaller values under-use accepted traces, while a larger value degrades both validation splits. We therefore use $\gamma = 0.05$ in the main experiments.

Table 9: Additional evaluations: Llama-3.1-8B general/cyber benchmarks and Qwen2.5-Coder-3B text-to-SQL transfer. Higher is better except WMDP-Cyber. Underlined values indicate the best score in each column.

**(a) Other benchmarks**

| Model | MMLU Pro ↑ | IFEval ↑ | Canary Exploit ↑ | WMDP Cyber ↓ |
|---|---|---|---|---|
| Llama-3.1-8B | 44.1 | 72.3 | 10.9 | 46.7 |
| Llama-Primus | 41.5 | 68.2 | 29.8 | 46.3 |
| Foundation-Sec | 42.5 | 67.1 | 0.0 | 45.3 |
| Llama-3.1-8B-GRPO | 45.6 | 72.3 | 0.0 | 47.1 |
| Llama-3.1-8B-MinervaRL | 43.6 | 73.8 | 28.4 | 47.1 |

**(b) Text-to-SQL transfer**

| Model | Spider Dev | Spider Test | BIRD Dev | DK | Syn | Real. | Live | Avg |
|---|---|---|---|---|---|---|---|---|
| Qwen2.5-Coder | 77.4 | 77.6 | 50.3 | 65.6 | 67.2 | 69.7 | 3.7 | 58.8 |
| SQL-R1-3B | 77.1 | 77.4 | 53.3 | 70.5 | 67.5 | 66.5 | 6.3 | 59.8 |
| GRPO | 77.5 | 78.5 | 54.2 | 69.0 | 67.9 | 68.7 | 6.3 | 60.3 |
| MinervaRL | 78.2 | 79.3 | 52.5 | 69.2 | 70.7 | 69.3 | 6.7 | 60.8 |

Table 10: Distillation scale $\gamma$ sensitivity.

| $\gamma$ | Minerva-Dev | Athena-Mini | Avg. |
|---|---|---|---|
| 0.01 | 60.3 | 61.3 | 60.8 |
| 0.02 | 63.1 | 63.3 | 63.2 |
| 0.05 | 63.2 | 63.3 | 63.3 |
| 0.10 | 60.1 | 58.1 | 59.1 |

## 8 Limitations and Future Work

Our study has several limitations. **Compute and ablations.** Because RLVR training is expensive, we only sweep the distillation learning-rate scale $\gamma$ and leave broader MinervaRL hyperparameter sweeps, including the distillation interval $I$, per-interval cap $M$, and EMA decay $\alpha$, for future work. **Data coverage.** Minerva-CTI and our evaluation suite are primarily English-centric; multilingual CTI, regional reporting styles, and non-English security artifacts remain important extensions. **Trace filtering.** MinervaRL uses lightweight heuristics and a TextCNN filter for ACR trace selection. Stronger LLM-based filters may improve trace quality but would increase cost, motivating more scalable quality estimators. **Training overhead.** MinervaRL adds overhead from ACR generation, log-probability computation, and periodic distillation. Appendix K.4 reports a 39.3% increase in a 10-step Llama-3.1-8B timing study, although Figure 5 shows faster progress toward strong validation performance. Reducing this overhead while preserving the support-seeding benefit is an important direction.

## 9 Conclusion

We presented **Minerva**, a verifiable-reward training framework for cyber threat intelligence LLMs. Minerva-CTI provides verifier-checkable tasks across vulnerability, detection, and procedure-oriented workflows, while MinervaRL extends GRPO with hardness-gated answer-conditioned rationale generation and original-prompt distillation. Across 12 CTI benchmarks and four open-weight backbones, MinervaRL improves the average score over matched base models, standard GRPO, controlled SFT/rejection-finetuning baselines, and an off-policy RL baseline. Task-level paired analysis shows the clearest gains on verifier-aligned structured tasks, while improvements also extend to SOC reasoning and extraction evaluations. These results show that CTI's structured taxonomies and canonical identifiers are well suited to verifier-driven post-training, and that support-seeding with filtered answer-conditioned traces can improve RLVR under sparse rewards.

**Data and Code Availability**

We will release the Minerva codebase, data-construction pipeline, evaluation tools, and derived Minerva-CTI splits for research use. Minerva-CTI is built from public CTI sources with reuse-compatible terms, including MITRE ATT&CK/CWE/CAPEC, CVE/NVD, Mappings Explorer, Sigma, Atomic Red Team, Microsoft Sentinel, and Splunk Security Content. Released artifacts will preserve required upstream notices, licenses, and attribution.

**Broader Impact Statement**

This work is intended to improve open-weight LLMs for defensive CTI analysis. Minerva-CTI is curated from public CTI and detection-engineering resources and targets analyst workflows such as indicator/entity extraction, vulnerability triage, ATT&CK/CWE/CVSS mapping, mitigation recommendation, and detection-rule interpretation. We do not train models for exploit generation, malware development, phishing, intrusion execution, or operational attack planning. Some evaluations, including exploit-related diagnostics, may nevertheless improve because stronger models better understand vulnerability semantics and attack-technique mappings; we treat such results as dual-use signals rather than deployment objectives. Potential misuse includes helping adversaries organize vulnerability knowledge, map attack techniques, prioritize targets, or automate threat analysis. We will therefore accompany released artifacts with defensive-use documentation, model/data cards, and cyber-safety review, and use staged release for model checkpoints or artifacts if review indicates elevated misuse risk. Finally, incorrect CWE labels, ATT&CK techniques, CVSS vectors, or mitigation recommendations can mislead downstream security decisions, so Minerva outputs should be treated as analyst-assistive signals rather than authoritative judgments.

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

# A Minerva-CTI Dataset

Minerva-CTI comprises 16 tasks covering vulnerability descriptions, detection content, and structured threat knowledge bases. Each task is defined as an input–target prediction problem derived from a specific upstream source, with targets expressed as canonical identifiers (e.g., ATT&CK technique, tactic, or mitigation IDs). Across all tasks, the dataset contains 32,000 training instances and 1,200 validation instances. Table 11 summarizes the per-task sample counts for the training and validation splits in Minerva-CTI.

**1. Mappings-Explorer CVE→ATT&CK Exploitation.** This task maps vulnerability descriptions to the ATT&CK technique directly used for exploitation. Given a CVE description as input, the model predicts a single ATT&CK technique identifier (formatted as `Txxxx` or `Txxxx.yyy`). Ground-truth labels are obtained from the Center for Threat-Informed Defense Mappings Explorer CVE→ATT&CK mappings, with technique definitions aligned to the ATT&CK catalog (Center for Threat-Informed Defense, 2026; MITRE Corporation, 2026d).

**2. Mappings-Explorer CVE→ATT&CK Primary Impact.** This task focuses on the main adversarial impact resulting from successful exploitation. The input is the CVE description, and the target is a single ATT&CK technique identifier corresponding to the primary post-exploitation effect (e.g., credential access or privilege escalation). Labels are sourced from the Mappings Explorer and normalized using the ATT&CK technique taxonomy (Center for Threat-Informed Defense, 2026; MITRE Corporation, 2026d).

**3. Mappings-Explorer CVE→ATT&CK Secondary Impact.** This task predicts a subsequent impact enabled by the primary impact. The input consists of the CVE description concatenated with the given primary-impact technique ID, and the target is a single ATT&CK technique identifier representing the secondary effect. Ground-truth annotations are derived from the Mappings Explorer and mapped to canonical ATT&CK identifiers (Center for Threat-Informed Defense, 2026; MITRE Corporation, 2026d).

**4. Sigma→ATT&CK Tactics.** This task infers high-level adversary intent from detection logic. Given a Sigma rule excerpt—including the rule title, `logsource`, and `detection` fields—the model predicts a multi-label set of ATT&CK tactic identifiers (formatted as `TA000x`). Sigma rules are sourced from the public Sigma repository, and annotations are expressed using the ATT&CK tactic taxonomy (SigmaHQ, 2026; MITRE Corporation, 2026d).

**5. Sigma→ATT&CK Technique.** This task maps detection logic to the specific adversarial behavior it is designed to identify. Using the same Sigma rule excerpt as input, the model predicts a single ATT&CK technique identifier (formatted as `Txxxx` or `Txxxx.yyy`). Rules are drawn from the Sigma repository, with targets aligned to canonical ATT&CK technique identifiers (SigmaHQ, 2026; MITRE Corporation, 2026d).

**6. Atomic Red Team→ATT&CK Technique.** This task maps adversary procedure descriptions to their corresponding ATT&CK techniques. The input is an Atomic Red Team procedure snippet that includes execution steps or commands and platform context, and the target is a single ATT&CK technique identifier. Examples are drawn from the Atomic Red Team repository, which is natively aligned with the ATT&CK framework (Red Canary, 2026; MITRE Corporation, 2026d).

**7. Microsoft Sentinel→ATT&CK Technique.** This task links analytics rules to the ATT&CK techniques they are intended to detect. The input comprises the rule title, description, and associated KQL query, and the target is a single ATT&CK technique identifier. Rules are sourced from the Microsoft Sentinel content repository, with labels normalized to the ATT&CK taxonomy (Microsoft, 2026; MITRE Corporation, 2026d).

**8. Splunk Security Content→ATT&CK Technique.** This task maps SPL-based detection content to the ATT&CK technique it targets. The input consists of an SPL query, its detection narrative, and metadata, and the target is a single ATT&CK technique identifier. Content is obtained from Splunk Security Content, with annotations expressed using canonical ATT&CK technique IDs (Splunk, 2026; MITRE Corporation, 2026d).

**9. ATT&CK Scenario→Technique.** This task identifies the ATT&CK technique that best corresponds to a described adversary scenario. The input is a scenario text derived from ATT&CK procedure examples, and the target is a single ATT&CK technique identifier. Scenarios and labels are sourced directly from the ATT&CK knowledge base and its machine-readable releases (MITRE Corporation, 2026d;a).

**10. ATT&CK Scenario→Tactics.** This task infers the adversary intent categories implied by a scenario. Given the same scenario text as input, the model predicts a multi-label set of ATT&CK tactic identifiers (`TA000x`) associated with the underlying behavior. Annotations are derived from the ATT&CK taxonomy and its structured releases (MITRE Corporation, 2026d;a).

**11. ATT&CK Scenario→Mitigations.** This task predicts mitigations relevant to the behaviors described in a scenario. The input is the ATT&CK scenario text, and the target is a multi-label set of ATT&CK mitigation identifiers (`Mxxxx`) associated with the corresponding techniques. Mitigation mappings are obtained from the ATT&CK knowledge base (MITRE Corporation, 2026d;a).

**12. NVD CVE→CWE.** This task maps vulnerability descriptions to their underlying weakness categories. The input is the CVE description text, and the target is a multi-label set of CWE identifiers (formatted as `CWE-xxx`). CVE records are obtained from the National Vulnerability Database, with labels aligned to the CWE taxonomy (Byers et al., 2022; MITRE Corporation, 2026c).

**13. NVD CVE→CVSS v3.1.** This task predicts the CVSS v3.1 base vector associated with a vulnerability. Given a CVE description as input, the model outputs the corresponding CVSS v3.1 base vector string (e.g., `CVSS:3.1/AV:N/...`). CVE entries are sourced from the National Vulnerability Database, and targets follow the official CVSS v3.1 specification (Byers et al., 2022; FIRST.org, Inc., 2019).

**14. ATT&CK Threat Actor Attribution.** This task performs threat actor attribution from observed behaviors expressed as procedure text. Examples are derived from MITRE ATT&CK Enterprise intrusion-set (group) entries by collecting each actor's *techniques used* relationships and extracting the associated procedure descriptions that characterize how the actor operates (MITRE Corporation, 2026d;a). Training instances are sampled from per-actor pools of procedures, with counts allocated roughly in proportion to available procedure coverage (e.g., binning by technique count and enforcing minimum coverage) to prevent a small number of well-documented actors from dominating the dataset. To reduce lexical leakage, prompts are anonymized by replacing explicit actor mentions with a generic placeholder (e.g., "A threat actor") and by generalizing other named entities by type (e.g., campaign, malware, tool) while preserving the remaining structure. The true actor name is retained only as the target label, and aliases are recorded in a lookup table for evaluation and reward scoring.

**15. CAPEC Example→CAPEC.** This task maps attack example narratives to the CAPEC attack pattern they exemplify. The input is a CAPEC example description, and the target is a single CAPEC identifier (formatted as `CAPEC-xxx`). Both example texts and pattern identifiers are sourced from the CAPEC catalog (MITRE Corporation, 2026b).

**16. CAPEC Example→CWE.** This task associates CAPEC attack examples with the underlying software weakness categories they exploit. The input is the same CAPEC example narrative, and the target is a multi-label set of CWE identifiers (formatted as `CWE-xxx`). Example descriptions are drawn from CAPEC, with labels aligned to the CWE taxonomy (MITRE Corporation, 2026b;c).

# B  Reward Functions for RLVR

Each Minerva-CTI task is paired with a deterministic verifier whose form depends on the task output type. Given a prompt $x$, a model completion $c$, and a ground-truth target $t$, the verifier first extracts a task-specific prediction $\hat{y}$ from $c$ and then assigns a reward $r \in [0, 1]$. Single-label tasks use exact identifier matching, hierarchical ATT&CK tasks use partial credit for base-technique matches, set-valued tasks use set-$F_1$, CVSS

Table 11: Minerva training datasets and split sizes.

| Dataset | Target | Train | Val |
|---|---|---|---|
| Mappings-Explorer CVE→ATT&CK Exploitation | ATT&CK technique ID | 245 | 20 |
| Mappings-Explorer CVE→ATT&CK Primary Impact | ATT&CK technique ID | 210 | 20 |
| Mappings-Explorer CVE→ATT&CK Secondary Impact | ATT&CK technique ID | 64 | 10 |
| Sigma→ATT&CK Tactics | ATT&CK tactic IDs | 950 | 50 |
| Sigma→ATT&CK Technique | ATT&CK technique ID | 950 | 50 |
| Atomic Red Team→ATT&CK Technique | ATT&CK technique ID | 950 | 50 |
| Microsoft Sentinel→ATT&CK Technique | ATT&CK technique ID | 950 | 50 |
| Splunk Security Content→ATT&CK Technique | ATT&CK technique ID | 280 | 20 |
| ATT&CK Scenario→Technique | ATT&CK technique ID | 7,780 | 220 |
| ATT&CK Scenario→Tactics | ATT&CK tactic IDs | 1,950 | 50 |
| ATT&CK Scenario→Mitigations | ATT&CK mitigation IDs | 7,780 | 220 |
| NVD CVE→CWE | CWE IDs | 6,696 | 220 |
| NVD CVE→CVSS v3.1 | CVSS v3.1 vector | 1,900 | 100 |
| ATT&CK Threat Actor Attribution | Threat actor name | 779 | 60 |
| CAPEC Example→CAPEC | CAPEC ID | 340 | 40 |
| CAPEC Example→CWE | CWE IDs | 176 | 20 |
| Total | – | 32,000 | 1,200 |

tasks use score-distance credit, and threat-actor attribution accepts known aliases. Thus, rewards measure task correctness after answer extraction and normalization rather than surface-form similarity.

**System prompt and answer extraction.** All Minerva-CTI training tasks use a shared system prompt that asks the model to reason step by step and place the final answer inside `\boxed{...}`:

> **Train System Prompt**
>
> You are given a cyber threat intelligence question. Solve it by reasoning step by step.
> Present the final answer clearly inside \boxed{}.

For reward computation, we prioritize the final `\boxed{...}` span when present. During training, extraction is intentionally strict: if no boxed span is found, we accept only explicit answer lines such as `Answer:` or `Final answer:`, and require either one unambiguous identifier for single-label tasks or a well-formed identifier set for multi-label tasks. During validation and testing, extraction is more permissive: if no boxed answer is present, we try answer-line parsing, tagged answer spans, and the last non-empty line, then apply task-specific regular expressions to recover candidate identifiers.

**Normalization.** Before scoring, predictions and targets are canonicalized. The function $\text{norm}_{\text{id}}(\cdot)$ trims whitespace and uppercases identifiers. For ATT&CK technique IDs, we additionally normalize sub-technique notation by zero-padding suffixes when needed, e.g., $\texttt{T1059.3} \mapsto \texttt{T1059.003}$. For threat-actor names, $\text{norm}_{\text{actor}}(\cdot)$ lowercases text, removes non-alphanumeric characters, and collapses repeated whitespace. Set-valued predictions are deduplicated after normalization.

**Single identifier exact match.** For tasks with a single canonical identifier, such as CAPEC Example→CAPEC, the extracted prediction $\hat{y}$ receives binary exact-match credit:

$$r = \mathbb{1}\left[\text{norm}_{\text{id}}(\hat{y}) = \text{norm}_{\text{id}}(t)\right].$$

**ATT&CK technique identifier.** For tasks whose target is an ATT&CK technique ID, including CVE→ATT&CK, scenario→technique, and detection→technique tasks, we give full credit for an exact normalized ID match and partial credit when the base technique matches but sub-technique specificity differs.

Let $b(\cdot)$ return the base technique ID, e.g., T1059, and let $s(\cdot) \in \{0, 1\}$ indicate whether the ID includes a sub-technique suffix. The reward is

$$r = \begin{cases} 1.0, & \text{norm}_{\text{id}}(\hat{y}) = \text{norm}_{\text{id}}(t), \\ 0.5, & b(\hat{y}) = b(t) \ \wedge \ \big(s(\hat{y}) = 1 \ \vee \ s(t) = 1\big), \\ 0.0, & \text{otherwise.} \end{cases}$$

This credit reflects the fact that upstream CTI sources sometimes annotate the same behavior at different levels of ATT&CK granularity.

**Multi-label identifier sets.** For set-valued targets, including scenario→tactics, scenario→mitigations, NVD CVE→CWE, and CAPEC Example→CWE, we normalize the predicted and target identifier sets as $P$ and $T$. Invalid identifiers are discarded and duplicates are removed. The reward is set-$F_1$:

$$r = \begin{cases} 1.0, & P = T = \varnothing, \\ 0.0, & (P = \varnothing) \oplus (T = \varnothing), \\ \dfrac{2|P \cap T|}{|P| + |T|}, & \text{otherwise,} \end{cases}$$

where $\oplus$ denotes exclusive-or. This penalizes both missing required labels and adding spurious labels.

**CVSS v3.1 base vector.** For NVD CVE→CVSS v3.1, the prediction must parse as a valid CVSS v3.1 base vector. We require each of the eight base metrics to appear exactly once:

$$\mathcal{M} = \{\text{AV}, \text{AC}, \text{PR}, \text{UI}, \text{S}, \text{C}, \text{I}, \text{A}\}.$$

Base metrics may appear in any order, and Temporal or Environmental metrics are ignored if present. If the prefix, version, metric set, or metric values are invalid, the reward is 0. Otherwise, we compute the CVSS v3.1 base score $s(\cdot) \in [0, 10]$ for the predicted and target vectors and assign distance-based credit:

$$r = \max\left(0, \ 1 - \frac{|s(\hat{y}) - s(t)|}{\delta}\right), \qquad \delta = 10.$$

**Threat actor attribution.** For ATT&CK threat-actor attribution, each target actor has an alias set $A(t)$ derived from ATT&CK group profiles, including the canonical name and known aliases. From the extracted answer span, we form a normalized candidate set $Y$ by splitting on commas and semicolons and applying $\text{norm}_{\text{actor}}(\cdot)$. The reward is

$$r = \mathbb{K}[Y \cap A(t) \neq \varnothing].$$

This gives credit when the model predicts either the canonical actor name or a recognized alias.

## C  Answer-Conditioned Rationale (ACR) Filtering

### C.1  ACR Generation

For hard prompts, MinervaRL generates answer-conditioned rationale (ACR) traces by augmenting the original training prompt with a label-revealing block. The ACR prompt provides the ground-truth label(s), optionally includes a truncated canonical reference, and asks the model to produce a short justification that ends in the same final-answer format required by the original task. The prompt explicitly discourages leakage language, such as stating that the answer was provided. Figure 6 shows the template.

### C.2  Candidate Filtering and Selection

For each buffered training prompt with unique identifier (UID) $i$, ACR generation produces $K$ candidate traces $\{c_{i,k}\}_{k=1}^{K}$. MinervaRL retains at most one trace per UID for self-distillation. Filtering proceeds in three stages.

---

**ACR Generation Prompt**

You are generating a reasoning trace for training.

GROUND_TRUTH_LABELS:
- <LABEL_1>
- <LABEL_2>

LABEL_REFERENCE:
<details_text>

Instructions:
- Write a short reasoning that would justify selecting the correct label(s) from the input.
- <TASK_OR_ENTITY_REASONING_HINT>
- Do NOT say or imply that the answer was provided (no phrases like "given the answer", "based on the provided label", "ground truth", etc.).
- End with the final answer in the same format required by the original task.

---

Figure 6: Answer-conditioned rationale (ACR) prompt template used to elicit training traces.

First, each candidate is scored by the task-specific Minerva verifier, yielding a correctness score $s_{i,k} \in [0,1]$; only candidates with $s_{i,k} = 1$ are considered for distillation. Second, verifier-correct candidates pass through heuristic filters that remove leakage and low-quality generations. Third, the remaining candidates are scored by a lightweight TextCNN quality classifier, and the highest-scoring eligible trace is selected.

**Heuristic filters.** We apply four heuristic checks before the ML filter. The leakage filter rejects candidates containing curated phrases or regex patterns that directly refer to provided labels, references, or ground-truth answers. The reasoning-length filter rejects candidates whose reasoning portion, excluding the final answer line, contains fewer than 100 characters. The grounding filter rejects candidates whose reasoning has Jaccard overlap below 0.05 with the task description plus label reference, computed over lowercased alphanumeric tokens after removing ID-like tokens. Finally, the degeneracy filter rejects repetitive responses when the response has at least 30 tokens. Degeneracy checks include $n$-gram repetition thresholds, repeated-window checks, and near-duplicate sentence checks. Specifically, we reject if $\mathrm{rep}_3 \geq 0.70$ or $\mathrm{rep}_4 \geq 0.75$; if repeated windows of size 24 and stride 12 are exact matches or have 3-gram Jaccard similarity at least 0.9; or if sentences within a 6-sentence window have SequenceMatcher similarity at least 0.75 when both sentences contain at least 6 words. These thresholds apply only to the heuristic repetition checks; the TextCNN acceptance threshold is $\tau_q = 0.5$.

**TextCNN quality filter.** Candidates that pass the heuristic filters are scored by the deployed response-only TextCNN classifier. The classifier tokenizes alphanumeric response tokens, truncates to at most 1024 tokens, and outputs a GOOD probability $q_{i,k} \in [0,1]$. We keep candidates with $q_{i,k} \geq \tau_q$, where $\tau_q = 0.5$. Let $L_{i,k}$ and $D_{i,k}$ denote whether candidate $k$ is rejected by leakage/quality heuristics or degeneracy heuristics, respectively. The eligible set is

$$\mathcal{E}_i = \{k \mid s_{i,k} = 1,\ L_{i,k} = 0,\ D_{i,k} = 0,\ q_{i,k} \geq \tau_q\}.$$

If $\mathcal{E}_i = \varnothing$, UID $i$ contributes no distillation example in that interval. Otherwise, we select the highest-confidence candidate,

$$k^\star = \arg\max_{k \in \mathcal{E}_i} q_{i,k},$$

breaking ties uniformly at random. The selected distillation record pairs the original answer-free prompt with the accepted ACR response, i.e., $(x_i, c_{i,k^\star})$.

## C.3 Filtering Dynamics

Figure 7 tracks the ACR filtering pipeline during training for Llama-3.1-8B-Instruct. We report the fraction of batches that pass the heuristic filter, pass the ML filter, and satisfy UID coverage, meaning that accepted traces span a sufficient number of distinct prompts. These rates increase over training, indicating that as the actor and EMA teacher improve, more ACR candidates are both verifier-correct and suitable for stable original-prompt distillation.

# D Training the TextCNN Reasoning-Quality Classifier

MinervaRL uses a lightweight TextCNN classifier to filter verifier-correct ACR traces before distillation. The classifier is trained to identify rationale quality rather than answer correctness: all training examples are already verifier-correct, and labels distinguish useful traces from traces with leakage, incoherence, unsupported reasoning, or other quality failures.

## D.1 Dataset Curation

We construct the TextCNN training data from Minerva-CTI prompts. For each prompt, we generate responses in two formats: *plain* responses from the original prompt, and *hinted* responses from an ACR-style prompt containing the ground-truth label, a label reference, and an explicit training-trace marker. Responses are generated with a diverse set of open or open-weight instruction-tuned LLMs using typical decoding settings of temperature 0.7 and maximum length 1024 new tokens.

To ensure that the classifier learns rationale quality rather than answer correctness, we first score all responses with the task verifier and retain only reward-correct responses with `reward` = 1. These responses are then labeled `GOOD` or `BAD` by a rubric-based LLM judge. The judge does not re-grade correctness; instead, it marks a response as `BAD` if it exhibits any quality failure, including leakage, incoherence, unsupported claims, mismatch between rationale and final answer, answer-only reasoning, refusals, prompt copying, or other artifacts. The judge is instructed to return a single JSON object, as shown in Figure 8.

To improve coverage of failure modes, we also synthesize additional `BAD` examples by conditioning a generator on randomly sampled rubric violations. These synthetic examples are retained only if they remain verifier-correct. We then build a balanced dataset by downsampling to 32k `GOOD` and 32k `BAD` examples, and split each class 80/20 into train and validation partitions.

## D.2 TextCNN Model and Training

The deployed classifier uses response text only; prompt text is not concatenated to the input. Responses are tokenized with the regex tokenizer `[A-Za-z0-9_]+`, lowercased, and truncated to a maximum of 1024 tokens. The vocabulary is built from the training split with `max_vocab`=100,000 and `min_freq`=2, using `<pad>` and `<unk>` as special tokens.

The model is a standard TextCNN: an embedding layer followed by 1D convolutions with kernel widths 3, 4, and 5; ReLU activations; max-over-time pooling; feature concatenation; dropout; and a two-way linear classifier. We train with AdamW and cross-entropy loss for 5 epochs using batch size 128. Hyperparameters are selected by grid search over learning rate $\{3\times10^{-4}, 6\times10^{-4}, 10^{-3}, 2\times10^{-3}, 4\times10^{-3}\}$, filters per kernel $\{256, 384\}$, dropout $\{0, 0.25\}$, embedding dimension $\{200, 300\}$, and maximum response-token budget. The deployed model uses learning rate $6\times10^{-4}$, 384 filters per kernel, embedding dimension 300, dropout 0.25, maximum response length 1024 tokens, and threshold 0.5. On the balanced validation split, it achieves 0.826 accuracy and $F_1 = 0.828$.

## D.3 Validation and Error Analysis

Table 12 reports the validation confusion matrix. False positives are `BAD` responses accepted by the TextCNN. As shown in Table 13, most false positives involve leakage: 738 of 1,206 false positives (61.2%) explicitly or implicitly reveal access to the provided answer or reference. Other common false-positive categories

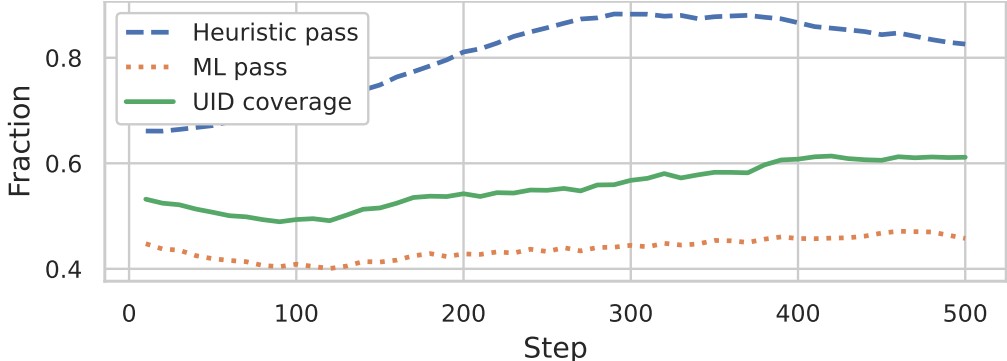

Figure 7: Acceptance rates of the ACR self-distillation pipeline over training for Llama-3.1-8B-Instruct: fraction of batches passing the heuristic filter, passing the machine-learning (ML) filter, and meeting unique identifier (UID) coverage.

Table 12: Validation confusion matrix for the TextCNN reasoning-quality filter at threshold 0.5.

|  | Predicted GOOD | Predicted BAD |
|---|---|---|
| Actual GOOD | 5,375 | 1,025 |
| Actual BAD | 1,206 | 5,194 |

are ungrounded claims (17.7%) and rationale-answer mismatch (13.2%). These errors are often fluent and plausible, which makes them difficult for a lightweight lexical classifier to reject.

A representative `scenario_to_attack_technique` false positive received $p(\text{GOOD}) = 0.829$ from TextCNN, but the judge labeled it `BAD`/Leakage because it implies access to the provided reference techniques:

> A malware has installed a bootkit on the system to maintain persistence. A bootkit modifies the boot sectors of a hard drive, allowing malicious code to execute before the operating system has loaded. This behavior aligns with the definition of techniques that involve modifying the operating system or firmware to achieve persistence.
> The Master Boot Record (MBR) and Volume Boot Record (VBR) are key components involved in the boot process, and modifying these areas would allow an adversary to execute malicious code before the operating system loads. This is a common method used by bootkits to persist on systems.
> *Considering the provided techniques*, the most suitable ID is `T1542.003`, which directly corresponds to Bootkit, as it describes the use of a bootkit to persist on systems. Therefore, the selected technique ID is `T1542.003`.

False negatives are `GOOD` responses rejected by the TextCNN. Table 14 shows that these are concentrated in open-ended ATT&CK generation tasks, especially `scenario_to_attack_mitigations` and `scenario_to_attack_technique`. These tasks permit diverse valid rationales, so a compact response-only classifier can reject acceptable traces whose style differs from common `GOOD` examples. This error profile is consistent with the component ablations in Table 5: removing the full filtering pipeline or only the ML filter reduces AthenaBench-Mini and average performance relative to full MinervaRL, despite occasional TextCNN errors.

## E   Evaluation Datasets

We evaluate on 12 CTI and cybersecurity benchmarks covering multiple-choice knowledge QA, SOC-style reasoning, structured taxonomy prediction, vulnerability assessment, and information extraction. Each benchmark is evaluated with task-specific answer extraction and the corresponding exact-match or structured metric described in Section 5.2. Meta tasks report the aggregate sample count over their constituent subtasks.

---

**LLM Judge Prompt**

You are an expert CTI evaluator. You will be given a Question and a Response. The Question may include answer/reference text as hints.

Label the Response as GOOD or BAD. Mark BAD if any single criterion below is present; otherwise mark GOOD. Do not grade correctness beyond these checks.

Return exactly one JSON object on a single line and nothing else.
- GOOD: {"label": "GOOD"}
- BAD: {"label": "BAD", "category_id": <number>, "category_title": "<title>"}

If multiple BAD criteria apply, choose the lowest-numbered category.

Criteria (one per bullet):
1. Leakage: says or implies the answer/label/options/reference were provided, or quotes/paraphrases provided reference text instead of reasoning from the prompt.
2. Incoherent: loops, repeated phrases/lines, templated filler, or gibberish.
3. Ungrounded: invents concrete details not in the question (extra CVEs, vendors, malware names, IOCs, dates, techniques, etc.).
4. Mismatch: reasoning supports a different label than the final answer, or directly contradicts it.
5. Unsupported: provides reasoning but does not use evidence from the prompt to justify the answer (hand-wavy or irrelevant justification), OR provides an answer with no reasoning at all (answer-only).
6. Other: refusals, policy/meta artifacts, generic CTI tutorials, or prompt copying.

Example outputs:
{"label": "GOOD"}
{"label": "BAD", "category_id": 1, "category_title": "Leakage"}

QUESTION:
{QUESTION}

RESPONSE:
{RESPONSE}

---

Figure 8: Prompt template used to label reward-correct responses as `GOOD`/`BAD` for training the TextCNN rationale-quality classifier.

Table 13: False positives by LLM-judge rubric category for the TextCNN filter at threshold 0.5.

| Rubric category | Count | Share of FP |
|---|---|---|
| Leakage | 738 | 61.2% |
| Ungrounded | 214 | 17.7% |
| Mismatch | 159 | 13.2% |
| Incoherent | 77 | 6.4% |
| Unsupported | 18 | 1.5% |

Table 14: False negatives by task for the TextCNN filter at threshold 0.5.

| Task | Count | Share of FN |
|---|---|---|
| scenario_to_attack_mitigations | 392 | 38.2% |
| scenario_to_attack_technique | 194 | 18.9% |
| cve_to_cwe | 173 | 16.9% |
| scenario_to_attack_tactics | 64 | 6.2% |
| sentinel_to_attack_technique | 35 | 3.4% |
| sigma_to_attack_technique | 35 | 3.4% |
| sigma_to_attack_tactics | 34 | 3.3% |
| cve_to_cvss_v31 | 28 | 2.7% |
| threat_actor | 23 | 2.2% |
| art_to_attack_technique | 20 | 2.0% |
| Other tasks | 27 | 2.6% |

Table 15: Evaluation datasets used for Table 1.

| Task | # Samples | Output format and evaluation target |
|---|---|---|
| CKT (Alam et al., 2025) | 3000 | Five-option CTI knowledge QA; output a single option A–E. |
| CyberMetric (Tihanyi et al., 2024) | 2000 | Four-option cybersecurity knowledge QA; output a single option A–D. |
| SOCEval (Deason et al., 2025) | 588 | Multi-select SOC-style reasoning over threat-intelligence reports; output a JSON object in `<json_object>` tags with a `correct_answers` array. |
| RCM (Alam et al., 2025) | 2000 | Root-cause mapping from CVE description to a single CWE identifier, e.g., `CWE-####`. |
| VSP (Alam et al., 2025) | 2000 | Vulnerability severity prediction; output a full CVSS v3.1 base vector, e.g., `CVSS:3.1/AV:N/....` |
| ATE (Alam et al., 2025) | 500 | ATT&CK technique extraction from an attack scenario; output a single technique or sub-technique ID, e.g., `T####` or `T####.###`. |
| RMS (Alam et al., 2025) | 500 | Risk mitigation recommendation; output a set of ATT&CK mitigation IDs, e.g., `M10xx`. |
| ElasticRule (Elastic, 2026) | 432 | Detection-rule mapping; map an Elastic rule and metadata to a single ATT&CK technique ID. |
| APTNER (Wang et al., 2022) | 1505 | APT-focused named-entity recognition; output a JSON object mapping entity types to extracted spans. |
| LANCE (Froudakis et al., 2025) | 466 | Indicator-of-compromise identification over candidate IPs, URLs, domains, and hashes; output candidate-level IoC labels. |
| AnnoCTR (Lange et al., 2024) | 1230 | STIX-style entity and relation extraction; meta task over entity extraction, entity typing, relation existence, and relation labeling with XML-like output tags. |
| AZERG (Lekssays et al., 2025) | 1333 | STIX-style entity and relation extraction; meta task over entity extraction, entity typing, relation existence, and relation labeling with XML-like output tags. |

# F   Statistical Uncertainty Analysis

**Bootstrap protocol.**   We use 2000 paired bootstrap resamples. For each task, backbone, and baseline comparison, examples are resampled within the task, both systems are evaluated on the same resampled examples, the original task metric is recomputed, and the percentile interval of MinervaRL minus the baseline is reported.

Table 16: Paired bootstrap 95% confidence intervals for MinervaRL minus each baseline on the 12-task macro average. The macro average is the unweighted mean over the 12 task metrics and is included for comparability with Table 1.

| Baseline | Llama-3.1-8B | Llama-3.2-3B | Qwen3-8B | Qwen3-4B |
|---|---|---|---|---|
| Base | +11.6 [10.8, 12.5] | +15.2 [14.5, 16.0] | +15.7 [14.5, 17.1] | +20.4 [19.0, 21.3] |
| STaR | +10.5 [9.7, 11.3] | +11.0 [10.3, 11.8] | +16.5 [15.4, 17.6] | +2.6 [1.5, 3.6] |
| DART | +6.6 [5.7, 7.5] | +4.3 [3.5, 5.1] | +10.5 [9.5, 11.8] | +8.6 [7.4, 9.7] |
| GRPO | +4.2 [3.5, 5.1] | +6.2 [5.3, 7.1] | +6.4 [5.6, 7.6] | +0.4 [-0.8, 1.3] |
| LUFFY | +3.3 [2.5, 4.1] | +2.9 [2.1, 4.0] | +0.1 [-0.6, 0.8] | +4.0 [3.0, 4.7] |

## F.1   Full Pairwise Confidence Intervals

Table 17: Per-task paired bootstrap 95% confidence intervals for MinervaRL minus Base, in percentage points.

| Task | Llama-3.1-8B | Llama-3.2-3B | Qwen3-8B | Qwen3-4B |
|---|---|---|---|---|
| CKT | +6.3 [4.6, 8.0] | -0.6 [-1.9, 0.7] | +6.0 [4.5, 7.5] | +24.4 [22.4, 26.5] |
| CyberMetric | +1.1 [-0.3, 2.5] | +1.1 [-0.2, 2.4] | +18.9 [16.8, 21.0] | +29.9 [27.2, 32.5] |
| SOCEval | -0.1 [-1.9, 1.9] | -1.1 [-3.1, 0.9] | +3.8 [2.2, 5.4] | +7.9 [5.6, 10.1] |
| RCM | +20.4 [18.3, 22.5] | +42.0 [40.0, 44.2] | +12.2 [10.5, 14.0] | +14.3 [12.7, 16.1] |
| VSP | +11.6 [10.6, 12.7] | +74.3 [73.2, 75.5] | +10.4 [9.4, 11.3] | +3.5 [2.6, 4.4] |
| ATE | +31.0 [26.8, 35.4] | +20.8 [17.4, 24.2] | +18.6 [14.8, 22.6] | +21.2 [17.6, 24.8] |
| RMS | +35.4 [32.1, 38.9] | +28.9 [26.3, 31.7] | +13.6 [10.7, 16.5] | -1.5 [-2.8, -0.2] |
| ElasticRule | +26.2 [21.1, 30.8] | +19.0 [15.0, 22.9] | +13.0 [9.0, 17.4] | +22.9 [19.0, 27.1] |
| APTNER | +0.6 [-0.3, 1.6] | -8.8 [-10.3, -7.2] | +6.1 [4.7, 7.6] | +26.8 [25.0, 28.7] |
| LANCE | +6.4 [2.2, 10.1] | +4.5 [-0.0, 9.5] | +29.8 [18.7, 44.5] | +38.7 [28.9, 48.6] |
| AnnoCTR | -0.4 [-2.7, 2.0] | +1.9 [-0.2, 3.9] | +33.2 [29.1, 35.8] | +33.8 [27.7, 36.1] |
| AZERG | +0.8 [-1.9, 3.6] | +0.8 [-1.5, 3.0] | +23.3 [20.6, 26.0] | +22.6 [18.0, 24.8] |

Table 18: Per-task paired bootstrap 95% confidence intervals for MinervaRL minus STaR-CTI, in percentage points.

| Task | Llama-3.1-8B | Llama-3.2-3B | Qwen3-8B | Qwen3-4B |
|---|---|---|---|---|
| CKT | +3.9 [2.2, 5.5] | +6.7 [5.1, 8.3] | +39.8 [37.7, 41.9] | -6.1 [-7.8, -4.5] |
| CyberMetric | +2.4 [1.0, 3.9] | +7.6 [5.8, 9.5] | +38.9 [36.5, 41.3] | -8.2 [-10.0, -6.5] |
| SOCEval | +3.0 [1.3, 4.9] | +4.3 [2.1, 6.4] | +2.4 [1.0, 3.9] | -2.1 [-4.4, 0.2] |
| RCM | +11.4 [9.7, 13.2] | +20.8 [19.0, 22.7] | +28.1 [26.1, 30.3] | +38.8 [36.6, 41.0] |
| VSP | +13.8 [12.8, 15.0] | +23.7 [22.0, 25.4] | +10.7 [9.8, 11.6] | -0.2 [-1.2, 0.7] |
| ATE | +19.2 [14.6, 23.8] | +17.6 [14.0, 21.0] | +16.4 [12.8, 20.0] | +19.6 [16.2, 23.2] |
| RMS | +32.5 [29.2, 36.0] | +22.3 [19.5, 25.4] | +16.6 [13.8, 19.5] | -1.5 [-2.7, -0.2] |
| ElasticRule | +19.2 [14.8, 23.8] | +18.3 [14.8, 22.2] | +17.1 [13.2, 21.1] | +15.3 [11.3, 19.2] |
| APTNER | +1.0 [-0.1, 2.1] | -4.7 [-6.1, -3.3] | -0.5 [-2.0, 1.0] | -3.8 [-5.5, -2.1] |
| LANCE | +4.6 [1.5, 7.5] | +5.9 [1.9, 10.7] | +19.2 [9.9, 29.0] | -5.3 [-14.2, 3.3] |
| AnnoCTR | +3.8 [1.3, 6.3] | +4.1 [1.7, 6.6] | +3.0 [-1.1, 6.2] | +3.5 [-0.8, 5.7] |
| AZERG | +11.6 [9.1, 14.0] | +5.7 [3.1, 8.1] | +6.7 [4.3, 9.1] | -18.6 [-21.2, -16.1] |

Table 19: Per-task paired bootstrap 95% confidence intervals for MinervaRL minus DART-CTI, in percentage points.

| Task | Llama-3.1-8B | Llama-3.2-3B | Qwen3-8B | Qwen3-4B |
|---|---|---|---|---|
| CKT | +2.6 [1.2, 4.1] | -2.0 [-3.2, -0.7] | +38.6 [36.6, 40.6] | +24.6 [22.5, 26.8] |
| CyberMetric | +1.9 [0.6, 3.1] | -0.1 [-1.4, 1.2] | +32.5 [30.1, 34.9] | +37.9 [35.3, 40.5] |
| SOCEval | +3.3 [1.4, 5.2] | +0.1 [-1.8, 1.9] | +2.4 [1.1, 3.8] | +4.0 [2.0, 6.1] |
| RCM | +4.7 [3.2, 6.3] | +2.3 [1.1, 3.7] | +17.7 [15.8, 19.6] | +5.2 [3.9, 6.7] |
| VSP | +14.6 [13.6, 15.7] | +15.9 [14.9, 17.0] | +6.7 [5.8, 7.6] | +6.2 [5.3, 7.2] |
| ATE | +10.6 [6.2, 15.2] | +5.0 [2.2, 7.6] | +18.0 [14.0, 22.0] | +4.8 [1.6, 8.2] |
| RMS | +14.4 [11.0, 18.1] | +11.5 [8.6, 14.5] | +13.6 [10.9, 16.3] | -0.8 [-2.0, 0.4] |
| ElasticRule | +9.5 [5.3, 13.7] | +8.6 [5.5, 11.6] | +5.6 [1.4, 10.0] | +10.4 [7.2, 13.9] |
| APTNER | +0.1 [-0.9, 1.1] | -5.8 [-7.2, -4.5] | -0.5 [-1.8, 0.8] | +0.4 [-1.2, 2.1] |
| LANCE | +9.8 [3.9, 16.3] | +8.4 [1.7, 14.8] | -1.5 [-10.8, 8.5] | +9.2 [-0.5, 21.2] |
| AnnoCTR | +3.2 [0.9, 5.6] | -0.8 [-3.0, 1.3] | -3.5 [-6.2, 0.1] | +7.1 [2.0, 9.3] |
| AZERG | +4.4 [2.0, 7.0] | +8.0 [5.8, 10.1] | -3.4 [-5.3, 0.2] | -6.1 [-10.5, -3.2] |

Table 20: Per-task paired bootstrap 95% confidence intervals for MinervaRL minus GRPO, in percentage points.

| Task | Llama-3.1-8B | Llama-3.2-3B | Qwen3-8B | Qwen3-4B |
|---|---|---|---|---|
| CKT | +2.0 [0.6, 3.5] | -0.2 [-1.6, 1.1] | +8.0 [6.5, 9.5] | -4.2 [-5.8, -2.6] |
| CyberMetric | -1.1 [-2.5, 0.3] | +0.3 [-1.1, 1.5] | +15.5 [13.5, 17.5] | -5.7 [-7.4, -4.0] |
| SOCEval | +1.7 [-0.4, 3.8] | -3.7 [-5.8, -1.4] | -1.4 [-2.7, -0.1] | +0.2 [-1.6, 2.1] |
| RCM | +2.4 [1.1, 3.8] | +8.3 [6.8, 9.8] | +4.3 [2.9, 5.8] | -0.9 [-2.1, 0.2] |
| VSP | +5.0 [4.3, 5.7] | +20.6 [18.9, 22.4] | +10.6 [9.7, 11.7] | -8.1 [-9.0, -7.3] |
| ATE | +16.4 [12.6, 20.4] | +16.6 [13.2, 20.2] | +8.4 [5.8, 11.4] | +5.2 [2.8, 7.6] |
| RMS | +11.2 [7.8, 14.4] | +13.9 [11.1, 16.9] | +12.0 [9.1, 14.8] | +1.3 [-0.7, 3.2] |
| ElasticRule | +8.3 [3.9, 12.7] | +14.8 [11.6, 18.3] | +3.9 [-0.2, 7.9] | +7.2 [3.7, 10.9] |
| APTNER | +1.4 [0.5, 2.4] | -10.8 [-12.3, -9.3] | -0.3 [-1.5, 1.0] | -4.4 [-5.9, -2.9] |
| LANCE | -1.5 [-5.7, 2.2] | +11.7 [2.7, 19.3] | +8.8 [0.8, 19.9] | -1.5 [-10.5, 9.3] |
| AnnoCTR | +0.7 [-1.6, 2.9] | -0.7 [-2.4, 1.1] | +5.2 [3.2, 8.1] | +10.7 [4.4, 12.9] |
| AZERG | +4.1 [2.3, 7.7] | +4.0 [1.5, 7.0] | +2.0 [0.1, 5.6] | +5.1 [0.3, 7.8] |

Table 21: Per-task paired bootstrap 95% confidence intervals for MinervaRL minus LUFFY-CTI, in percentage points.

| Task | Llama-3.1-8B | Llama-3.2-3B | Qwen3-8B | Qwen3-4B |
|---|---|---|---|---|
| CKT | +0.0 [-1.4, 1.3] | +0.4 [-0.8, 1.5] | -0.9 [-1.9, 0.2] | +16.0 [13.9, 18.0] |
| CyberMetric | +0.6 [-0.7, 1.8] | -0.4 [-1.6, 0.8] | -0.1 [-1.2, 1.0] | +23.6 [21.1, 25.9] |
| SOCEval | +1.5 [-0.0, 2.9] | -0.8 [-2.4, 0.9] | +2.7 [1.0, 4.4] | +3.1 [1.1, 5.1] |
| RCM | +4.9 [3.7, 6.2] | +13.4 [11.9, 14.9] | -0.5 [-1.6, 0.6] | -0.0 [-1.2, 1.0] |
| VSP | +8.0 [7.2, 8.8] | +5.5 [4.5, 6.6] | +6.2 [5.5, 7.0] | +7.4 [6.5, 8.2] |
| ATE | +7.8 [4.2, 11.6] | +2.4 [-0.4, 5.4] | +2.6 [0.0, 5.2] | +7.8 [4.6, 11.0] |
| RMS | +7.8 [4.4, 11.2] | +5.1 [2.3, 8.0] | -5.7 [-8.5, -3.0] | -5.2 [-6.9, -3.6] |
| ElasticRule | +6.7 [2.5, 10.6] | -0.2 [-3.0, 2.5] | -10.4 [-14.1, -6.5] | +4.2 [0.7, 7.6] |
| APTNER | -1.3 [-2.1, -0.5] | +1.7 [0.3, 3.0] | +3.1 [2.0, 4.2] | -1.1 [-2.7, 0.5] |
| LANCE | +2.2 [-3.1, 7.3] | +14.2 [5.2, 22.2] | +0.4 [-5.4, 5.2] | -3.7 [-10.5, 4.4] |
| AnnoCTR | +0.6 [-2.1, 3.1] | -8.1 [-10.2, -3.7] | -2.1 [-5.5, 0.2] | +3.7 [-0.7, 5.8] |
| AZERG | +0.5 [-2.2, 3.4] | +1.6 [-0.9, 5.1] | +6.4 [3.8, 8.8] | -7.1 [-10.1, -4.9] |

### F.2 Rollout-Aware Confidence Intervals

Tables 22 to 25 report per-task paired bootstrap intervals for the held-out rollout-aware comparisons in Section 7.2. Each cell is MinervaRL minus GRPO in percentage points under the corresponding best-of-$k$ score.

Table 22: Per-task rollout-aware paired bootstrap 95% confidence intervals for MinervaRL minus GRPO at best-of-1, in percentage points.

| Task | Llama-3.1-8B | Llama-3.2-3B | Qwen3-8B | Qwen3-4B |
|---|---|---|---|---|
| CKT | +0.0 [-1.5, 1.6] | -0.4 [-2.0, 1.2] | +7.0 [5.3, 8.7] | +9.4 [7.6, 11.3] |
| CyberMetric | -0.5 [-1.9, 0.8] | +1.3 [0.0, 2.6] | +2.8 [1.4, 4.3] | +28.6 [26.0, 31.1] |
| SOCEval | +2.2 [0.0, 4.3] | -5.1 [-7.4, -3.1] | -2.3 [-4.6, -0.2] | -0.3 [-2.5, 1.8] |
| RCM | +0.7 [-0.8, 2.2] | +7.5 [6.0, 8.9] | +4.0 [2.6, 5.4] | -0.7 [-1.9, 0.6] |
| VSP | +5.0 [4.3, 5.7] | +20.0 [18.4, 21.8] | +9.3 [8.4, 10.2] | -8.2 [-9.0, -7.3] |
| ATE | +16.0 [12.0, 19.8] | +16.8 [13.4, 20.4] | +8.6 [5.6, 11.8] | +3.8 [1.0, 6.6] |
| RMS | +7.6 [4.3, 11.1] | +11.2 [8.2, 14.1] | +10.5 [7.3, 13.4] | -4.3 [-6.8, -1.8] |
| ElasticRule | +8.6 [4.4, 12.7] | +15.0 [11.3, 18.8] | +0.2 [-3.2, 3.7] | +4.6 [1.2, 8.3] |
| APTNER | -0.1 [-1.1, 0.8] | -11.7 [-13.0, -10.6] | -3.2 [-5.0, -1.4] | +10.3 [8.4, 12.3] |
| LANCE | -5.5 [-10.0, -1.6] | +10.2 [2.2, 18.6] | +2.8 [-7.0, 15.2] | +12.0 [0.0, 24.5] |
| AnnoCTR | -5.3 [-7.8, -1.0] | +5.6 [3.3, 7.5] | +0.2 [-2.7, 5.4] | -1.4 [-3.7, 3.0] |
| AZERG | +3.2 [1.0, 6.5] | +18.0 [15.4, 20.5] | -6.2 [-9.0, -3.3] | -4.6 [-7.1, -2.0] |

Table 23: Per-task rollout-aware paired bootstrap 95% confidence intervals for MinervaRL minus GRPO at best-of-2, in percentage points.

| Task | Llama-3.1-8B | Llama-3.2-3B | Qwen3-8B | Qwen3-4B |
|---|---|---|---|---|
| CKT | +5.1 [3.7, 6.5] | +3.1 [1.8, 4.6] | +0.8 [-0.4, 2.1] | +1.0 [-0.4, 2.4] |
| CyberMetric | +0.7 [-0.5, 1.8] | +2.0 [0.8, 3.1] | +0.3 [-0.6, 1.4] | -0.3 [-1.7, 1.1] |
| SOCEval | +3.5 [1.4, 5.6] | -3.0 [-5.0, -0.9] | -0.7 [-2.4, 1.0] | +0.2 [-1.7, 2.1] |
| RCM | +3.1 [1.6, 4.6] | +9.1 [7.6, 10.7] | +5.5 [4.0, 7.1] | +1.0 [-0.1, 2.1] |
| VSP | +5.9 [5.3, 6.6] | +11.9 [10.6, 13.2] | +8.8 [7.9, 9.7] | -7.5 [-8.2, -6.6] |
| ATE | +19.4 [15.6, 23.2] | +20.2 [16.6, 24.2] | +8.6 [5.6, 11.8] | +4.6 [1.6, 7.4] |
| RMS | +13.2 [9.9, 16.6] | +11.2 [8.2, 14.4] | +16.9 [14.2, 19.7] | -5.3 [-8.1, -2.6] |
| ElasticRule | +12.7 [8.6, 17.1] | +14.6 [11.1, 18.3] | +4.2 [1.4, 6.9] | +4.2 [0.9, 7.4] |
| APTNER | +0.1 [-0.7, 1.0] | -15.7 [-16.9, -14.4] | -1.5 [-2.9, -0.2] | +5.0 [3.5, 6.5] |
| LANCE | -2.7 [-4.7, -0.7] | +5.7 [0.2, 10.7] | +4.6 [-3.8, 12.3] | +4.8 [-4.0, 12.6] |
| AnnoCTR | -2.8 [-5.4, 1.5] | +4.1 [2.9, 6.7] | +0.5 [-0.9, 4.1] | +4.6 [2.5, 10.8] |
| AZERG | -2.0 [-4.9, 1.5] | +4.4 [2.4, 6.5] | -1.3 [-4.0, 1.3] | -1.1 [-3.4, 1.3] |

Table 24: Per-task rollout-aware paired bootstrap 95% confidence intervals for MinervaRL minus GRPO at best-of-4, in percentage points.

| Task | Llama-3.1-8B | Llama-3.2-3B | Qwen3-8B | Qwen3-4B |
|---|---|---|---|---|
| CKT | +8.1 [6.8, 9.5] | +4.2 [3.1, 5.3] | -0.3 [-1.4, 0.8] | +0.6 [-0.6, 1.8] |
| CyberMetric | +1.1 [0.1, 2.1] | +2.3 [1.2, 3.4] | -0.4 [-1.3, 0.4] | -0.8 [-1.8, 0.2] |
| SOCEval | +6.9 [4.8, 9.2] | -0.6 [-2.6, 1.6] | +0.2 [-1.2, 1.6] | +0.6 [-1.2, 2.2] |
| RCM | +4.5 [3.1, 6.0] | +10.5 [9.0, 12.2] | +7.2 [5.8, 8.7] | +2.3 [1.2, 3.3] |
| VSP | +6.2 [5.6, 6.8] | +7.0 [6.1, 7.9] | +4.4 [3.6, 5.2] | -5.8 [-6.6, -5.1] |
| ATE | +20.4 [16.4, 24.2] | +23.2 [19.6, 27.0] | +10.0 [6.8, 13.4] | +7.0 [4.0, 10.0] |
| RMS | +17.9 [14.6, 21.2] | +12.7 [9.5, 15.7] | +22.3 [19.3, 25.5] | -3.1 [-5.8, -0.3] |
| ElasticRule | +16.4 [12.3, 20.6] | +16.0 [12.5, 19.4] | +5.6 [2.8, 8.6] | +5.1 [1.9, 8.3] |
| APTNER | +1.4 [0.5, 2.2] | -16.3 [-17.6, -15.0] | -2.1 [-3.4, -0.8] | +4.8 [3.3, 6.2] |
| LANCE | -1.5 [-3.3, 0.6] | +6.2 [0.8, 10.8] | +2.0 [-6.5, 10.3] | +1.3 [-4.4, 8.0] |
| AnnoCTR | -1.4 [-3.6, 2.9] | +5.5 [4.0, 8.2] | +2.3 [0.9, 4.3] | +4.2 [2.3, 10.3] |
| AZERG | +1.1 [-2.3, 5.6] | +7.8 [5.6, 9.9] | +2.5 [-0.0, 4.9] | +5.9 [4.0, 7.8] |

Table 25: Per-task rollout-aware paired bootstrap 95% confidence intervals for MinervaRL minus GRPO at best-of-8, in percentage points.

| Task | Llama-3.1-8B | Llama-3.2-3B | Qwen3-8B | Qwen3-4B |
|---|---|---|---|---|
| CKT | +9.4 [8.1, 10.7] | +6.6 [5.7, 7.7] | -1.7 [-2.6, -0.8] | +2.0 [1.0, 2.9] |
| CyberMetric | +1.9 [1.0, 2.9] | +3.1 [2.0, 4.2] | -0.7 [-1.5, -0.0] | +0.3 [-0.5, 1.0] |
| SOCEval | +8.5 [6.4, 10.6] | +1.0 [-1.1, 3.1] | +1.1 [-0.1, 2.2] | +0.5 [-1.1, 2.1] |
| RCM | +5.6 [4.1, 7.0] | +12.5 [10.9, 14.2] | +8.7 [7.2, 10.1] | +2.6 [1.5, 3.6] |
| VSP | +6.3 [5.7, 6.8] | +4.6 [4.0, 5.3] | -3.2 [-3.9, -2.6] | -0.9 [-1.5, -0.3] |
| ATE | +23.6 [19.8, 27.8] | +25.4 [21.4, 29.6] | +11.6 [8.2, 14.8] | +10.2 [7.2, 13.6] |
| RMS | +22.4 [19.1, 25.6] | +15.3 [12.5, 18.2] | +26.7 [23.7, 29.5] | +0.6 [-2.3, 3.7] |
| ElasticRule | +18.1 [13.9, 22.2] | +17.8 [13.9, 21.8] | +8.3 [5.3, 11.3] | +6.5 [3.5, 9.7] |
| APTNER | +1.8 [1.0, 2.6] | -15.7 [-17.0, -14.4] | -1.3 [-2.6, -0.1] | +5.3 [3.9, 6.7] |
| LANCE | -1.2 [-3.4, 1.3] | +2.6 [-0.6, 5.3] | -0.5 [-3.9, 2.3] | -1.9 [-4.9, 2.4] |
| AnnoCTR | +2.7 [-1.5, 6.8] | +7.3 [5.7, 10.2] | +1.5 [0.1, 3.0] | +8.0 [5.1, 10.5] |
| AZERG | +2.5 [-1.0, 6.4] | +7.8 [5.8, 9.8] | +3.7 [1.2, 6.1] | +4.5 [2.8, 6.3] |

## G  Additional Result Tables

### G.1  Rollout-Aware Task Scores

Table 26: Full rollout-aware best-of-$k$ task scores for GRPO and MinervaRL. Each task cell reports GRPO/MinervaRL in percent for the specified backbone and best-of-$k$ budget.

| Backbone | $k$ | CKT | CyberMetric | SOCEval | RCM | VSP | ATE | RMS | ElasticRule | APTNER | LANCE | AnnoCTR | AZERG |
|---|---|---|---|---|---|---|---|---|---|---|---|---|---|
| Llama-3.1-8B | 1 | 71.3/71.4 | 85.7/85.2 | 63.2/65.4 | 66.5/67.2 | 82.4/87.4 | 30.8/46.8 | 31.3/38.9 | 31.7/40.3 | 32.6/32.5 | 87.0/81.6 | 52.9/47.6 | 39.3/42.5 |
| Llama-3.1-8B | 2 | 75.4/80.5 | 89.0/89.7 | 65.8/69.3 | 67.2/70.3 | 83.9/89.9 | 32.2/51.6 | 33.8/47.1 | 32.4/45.1 | 36.0/36.2 | 88.0/85.3 | 55.7/52.9 | 47.8/45.8 |
| Llama-3.1-8B | 4 | 77.9/86.0 | 91.4/92.5 | 69.7/76.7 | 68.4/72.9 | 85.1/91.3 | 35.0/50.5 | 35.0/52.9 | 34.0/50.5 | 38.6/40.0 | 89.2/87.8 | 58.8/57.4 | 49.7/50.8 |
| Llama-3.1-8B | 8 | 80.0/89.4 | 92.9/94.8 | 71.6/80.1 | 69.2/74.8 | 86.0/92.3 | 35.4/59.0 | 36.0/58.4 | 34.3/52.3 | 41.0/42.8 | 90.1/88.8 | 61.8/64.5 | 51.4/53.9 |
| Llama-3.2-3B | 1 | 68.6/68.2 | 77.7/79.0 | 60.5/55.4 | 47.9/55.5 | 54.6/74.6 | 5.4/22.2 | 17.1/28.2 | 5.6/20.6 | 26.8/15.1 | 66.2/76.5 | 36.9/42.5 | 25.0/43.0 |
| Llama-3.2-3B | 2 | 73.1/76.2 | 80.5/82.5 | 63.0/60.0 | 48.9/58.0 | 70.0/81.9 | 5.6/25.8 | 22.3/33.4 | 6.5/21.1 | 30.7/15.1 | 77.6/83.3 | 43.5/47.6 | 47.9/52.3 |
| Llama-3.2-3B | 4 | 81.6/85.7 | 83.9/86.2 | 67.2/66.6 | 49.5/60.1 | 79.8/86.8 | 5.8/29.0 | 24.5/37.2 | 7.2/23.1 | 34.0/17.7 | 81.8/88.1 | 46.7/52.1 | 51.1/58.9 |
| Llama-3.2-3B | 8 | 82.9/89.5 | 85.7/88.8 | 68.8/69.8 | 50.3/62.8 | 85.1/89.7 | 5.8/31.2 | 26.3/41.6 | 7.9/25.7 | 37.0/21.3 | 87.1/89.7 | 51.6/58.9 | 55.2/63.0 |
| Qwen3-8B | 1 | 69.0/76.0 | 84.5/87.4 | 68.4/66.1 | 60.0/64.0 | 69.5/78.8 | 22.6/31.2 | 7.9/18.4 | 24.1/24.3 | 36.2/33.1 | 66.4/69.2 | 45.1/45.3 | 38.4/32.3 |
| Qwen3-8B | 2 | 82.3/83.1 | 92.1/92.5 | 74.7/74.0 | 61.5/67.0 | 73.8/82.6 | 24.6/33.2 | 8.3/25.2 | 26.6/30.8 | 45.1/43.6 | 74.7/79.3 | 53.5/53.9 | 49.2/47.9 |
| Qwen3-8B | 4 | 88.2/87.8 | 94.9/94.5 | 80.9/81.1 | 62.3/69.5 | 81.6/86.0 | 25.2/35.3 | 8.6/31.0 | 28.5/34.0 | 51.7/49.6 | 83.2/85.2 | 56.8/59.1 | 53.4/55.9 |
| Qwen3-8B | 8 | 92.8/91.1 | 96.8/96.0 | 83.7/84.8 | 62.8/71.5 | 91.6/88.4 | 25.8/37.4 | 9.0/35.7 | 29.4/37.7 | 55.3/53.9 | 92.3/91.8 | 59.5/60.9 | 58.1/61.8 |
| Qwen3-4B | 1 | 62.7/72.2 | 54.1/82.8 | 62.1/61.8 | 60.2/59.5 | 88.1/80.0 | 20.6/24.4 | 12.2/7.9 | 22.0/26.6 | 21.8/32.2 | 45.4/57.4 | 44.8/43.4 | 38.5/33.9 |
| Qwen3-4B | 2 | 80.4/81.4 | 90.6/90.3 | 69.2/69.5 | 61.5/62.5 | 89.6/82.1 | 22.0/26.6 | 14.7/9.4 | 25.0/29.2 | 34.4/39.3 | 64.2/69.0 | 54.3/58.9 | 48.6/47.5 |
| Qwen3-4B | 4 | 86.3/86.9 | 95.0/94.2 | 75.2/75.8 | 62.2/64.5 | 90.1/84.3 | 22.4/29.4 | 15.1/12.0 | 25.7/30.8 | 41.3/46.0 | 76.7/78.1 | 59.2/63.4 | 53.5/59.4 |
| Qwen3-4B | 8 | 89.9/91.9 | 96.3/96.5 | 80.0/80.5 | 62.8/65.4 | 90.8/89.9 | 22.4/32.6 | 15.9/16.5 | 26.4/32.9 | 46.0/51.3 | 89.7/87.8 | 65.6/73.5 | 58.4/62.9 |

### G.2  Component Ablation Task Scores

Table 27: Full Llama-3.1-8B component ablation scores on the 12 CTI evaluation tasks. All entries are task scores in percent.

| Ablation | CKT | CyberMetric | SOCEval | RCM | VSP | ATE | RMS | ElasticRule | APTNER | LANCE | AnnoCTR | AZERG | Avg. |
|---|---|---|---|---|---|---|---|---|---|---|---|---|---|
| GRPO | 71.9 | 85.4 | 63.0 | 66.3 | 82.6 | 32.0 | 30.9 | 32.2 | 32.7 | 86.2 | 49.6 | 39.5 | 56.0 |
| GRPO 12 rollouts | 71.2 | 85.0 | 64.6 | 66.8 | 76.3 | 31.4 | 17.0 | 31.7 | 36.3 | 82.6 | 48.8 | 35.4 | 53.9 |
| Answer-only SFT | 76.3 | 85.9 | 65.2 | 64.7 | 64.8 | 36.6 | 17.2 | 34.7 | 40.7 | 56.5 | 45.3 | 43.2 | 52.6 |
| In-loop answer SFT | 75.6 | 84.5 | 66.8 | 65.8 | 79.2 | 38.4 | 28.6 | 33.3 | 38.7 | 78.1 | 53.9 | 50.5 | 57.8 |
| No EMA teacher | 72.5 | 83.7 | 62.4 | 65.3 | 85.9 | 42.2 | 43.0 | 39.8 | 33.1 | 75.1 | 43.7 | 40.2 | 57.2 |
| No TextCNN filter | 71.2 | 85.3 | 63.6 | 64.0 | 78.6 | 45.0 | 36.5 | 41.4 | 36.0 | 82.5 | 47.5 | 41.1 | 57.7 |
| No filtering | 73.1 | 84.3 | 63.3 | 67.2 | 83.9 | 40.0 | 37.2 | 38.2 | 34.2 | 81.2 | 45.9 | 44.6 | 57.8 |
| Full MinervaRL | 73.9 | 84.2 | 64.7 | 68.8 | 87.6 | 48.4 | 42.1 | 40.5 | 34.1 | 84.6 | 50.3 | 43.7 | 60.2 |

## H  Additional Baseline Construction Details

This appendix describes the three additional training-method baselines reported in Table 1. All baselines use the same 32,000-example Minerva-CTI train split, the same task verifiers, and the same 12-task evaluation protocol as MinervaRL unless stated otherwise. Training and sampling hyperparameters for these baselines are listed with the other implementation settings in Appendix K.

### H.1 STaR-CTI

STaR-CTI adapts STaR (Zelikman et al., 2022) to verifier-scored CTI tasks. For each training example $(x, y^\star)$ in a round, we generate one *original* trace from the original prompt $x$ and one *rationalization* trace from a prompt that additionally reveals the gold answer $y^\star$. Both traces are scored by the Minerva verifier. We keep the original trace if it is verifier-correct; otherwise, we keep the rationalization trace if it is verifier-correct; otherwise, the example contributes no trace in that round.

The selected traces form that round's SFT dataset. We train a fresh copy of the base model for each round rather than continuing from the previous SFT checkpoint. The resulting checkpoint is then used as the generator for the next round. Checkpoints are selected with the same validation criterion used for the RL baselines: the average of Minerva-Dev and AthenaBench-Mini validation performance. The validation suite contains 1,200 Minerva-Dev examples and 950 Athena CTI examples covering ATE, CKT, RCM, RMS, TAA, and VSP.

### H.2 DART-CTI

DART-CTI adapts DART-Math (Tong et al., 2024) as a fixed rejection-finetuning baseline. We build a teacher-generated trace corpus once, then train each student model with SFT on that fixed corpus. The target corpus contains two accepted traces per Minerva-CTI training prompt, for 64,000 traces total.

The final corpus is built in four stages. First, we run plain rejection sampling from the original prompt, with a target of two accepted traces per prompt and a cap of 32 attempts per prompt. This produces 16,807 accepted plain traces from 880,782 generated attempts. Second, for prompts still missing traces, we use answer-guided generation with the same ACR-style prompt used by MinervaRL and accept traces that pass the verifier plus the heuristic/TextCNN filters; this adds 38,868 traces. Third, for the remaining hard tail, we keep answer-guided generation but disable the heuristic/TextCNN filters and require only verifier success; this adds 8,296 traces. Finally, 29 remaining missing traces are filled with a deterministic boxed gold-answer completion that is still checked by the verifier. The final training corpus therefore contains 16,807 plain traces, 47,164 guided-fill traces, and 29 direct-answer tail traces.

### H.3 LUFFY-CTI

LUFFY-CTI adapts LUFFY (Yan et al., 2025), an off-policy RLVR method that uses precomputed correct traces as off-policy guidance during RL training. We derive the off-policy guidance corpus from the DART-CTI trace corpus by selecting one accepted trace per training prompt, giving a 32,000-trace guidance set aligned with the Minerva-CTI train split.

We run LUFFY-CTI on the same four backbones used for GRPO and MinervaRL. This baseline is useful because it tests whether an off-policy trace-guided RL method closes the reward-sparsity gap without MinervaRL's online answer-conditioned rationale generation and periodic distillation. A practical distinction is that LUFFY-CTI requires the guidance corpus before RL begins: in our setup, constructing the 32,000-prompt trace corpus required 1,017,847 teacher attempts, or 31.8 attempts per prompt on average. By contrast, under the default MinervaRL schedule each prompt is seen about twice on average, with eight on-policy rollouts per visit and up to four additional ACR generations only for hard prompts, giving an upper bound of 24 generations per prompt across the full RL process.

## I Response-Quality Evaluation Details

This appendix describes the response-quality evaluation summarized in Figure 3. The goal is to complement verifier-based task metrics with a separate assessment of analyst-facing response quality, including readability, evidence use, and CTI concept precision.

### I.1 Evaluation Scope and Rubric

For each backbone family, we evaluate a shared set of 350 prompts: 50 prompts each from CKT, CyberMetric, RCM, VSP, ATE, RMS, and ElasticRule (Elastic, 2026). These tasks require justification or nontrivial CTI reasoning, making them suitable for prose-quality assessment. We exclude schema-heavy extraction and tagging tasks because their responses are dominated by format compliance and label recovery rather than explanatory quality.

Within each backbone family, we compare five variants: the base model, GRPO, MinervaRL, STaR-CTI, and DART-CTI. Model identities are hidden from the judge. GPT-5.2 scores each prompt-response pair independently using the three-criterion rubric in Figure 9. Each criterion is scored from 1 to 4, and the total response-quality score is the sum of writing quality, evidence use, and CTI concept use, giving a range of 3–12.

### I.2 Pairwise Preference Construction

The heatmaps in Figure 3 are derived from the pointwise rubric scores. For each prompt and each unordered pair of models within the same backbone family, we compare the two total response-quality scores. The model with the higher total score receives one pairwise win; if the scores are equal, the outcome is counted as a tie. Each heatmap cell reports the row model's win count divided by the number of shared prompts, with ties retained in the denominator rather than discarded.

### I.3 Human and Cross-Judge Agreement

We validate GPT-5.2 on a 100-example subset covering the five Llama-3.1-8B-family variants. Two human annotators independently score the same blind prompt-response items using the rubric in Figure 9. We also score the same subset with Claude Sonnet 4.6 as a cross-judge comparison. Agreement is measured on the total rubric score using quadratic weighted kappa (QWK) and Spearman correlation.

GPT-5.2 agrees with the human annotator average at a level close to human–human agreement, and Claude Sonnet 4.6 shows strong agreement with GPT-5.2. These results support using GPT-5.2 as a scalable response-quality judge for the full preference study, while treating the preference heatmaps as complementary to the verifier-based task metrics.

## J Additional Support for MinervaRL

### J.1 Target-Level Reward Sparsity

To characterize the sparsity that motivates hardness-gated answer-conditioned rationale generation, we analyze the DART-CTI stage-1 plain rejection-sampling traces with a maximum budget of 32 attempts per prompt. Figure 10 plots target-level aggregates for ATE and RCM. Question-weighted summaries from the target-level statistics show that ATE is sparser than RCM: ATE requires 26.7 attempts per question on average, yields 0.66 verifier-successful responses, and reaches the 32-attempt cap for 73.2% of questions, compared with 23.6 attempts, 0.86 verifier-successful responses, and a 61.5% cap rate for RCM. Both tasks also exhibit long target-level tails, with many identifiers clustered near the attempt cap and below one successful response per question.

### J.2 Why MinervaRL Can Expand Empirical Support

**Answer-level view.** Consider a training instance $(x, a^\star)$, where $x \in \mathcal{X}$ is the original prompt and $a^\star \in \mathcal{A}$ is the ground-truth structured answer, such as an ATT&CK technique ID, a set of mitigation IDs, or a CVSS vector. Let $g : \mathcal{Y} \to \mathcal{A}$ denote the task-specific extractor that maps a model output sequence $y \in \mathcal{Y}$ to its final extracted answer. For this analysis, we focus on the event of full verification and write

$$S(x, y; a^\star) := \mathbb{1}[g(y) = a^\star].$$
(13)

---

**Pointwise Response-Quality Judge Prompt**

You are an expert cyber threat intelligence (CTI) analyst evaluating the quality of a single model response. Your job is to score the response using the rubric below.

Rubric: use a 1–4 scale for each criterion, where 1 = poor, 2 = fair, 3 = good, and 4 = excellent.

1. **Writing quality.** Is the response easy to read and efficiently structured?
   **1:** no rationale, or hard to follow because of rambling, disorganization, or awkward phrasing.
   **2:** understandable, but noticeably wordy, repetitive, or clunky.
   **3:** clear and easy to follow, with minor issues in concision or structure.
   **4:** very clear, concise, well structured, and easy to scan.

2. **Evidence use.** Does the response justify its answer using details from the prompt?
   **1:** mostly asserts the answer with little or no prompt-based support.
   **2:** uses some prompt evidence, but the justification is weak, generic, or incomplete.
   **3:** supports the main answer with relevant prompt details, with only minor gaps.
   **4:** directly justifies the answer using strong and specific prompt evidence.

3. **CTI concept use.** Does the response use the right CTI concepts correctly?
   **1:** uses no relevant CTI concepts, or uses irrelevant concepts.
   **2:** uses some relevant CTI concepts, but with important mistakes or vague distinctions.
   **3:** mostly uses the right CTI concepts correctly, with minor imprecision.
   **4:** uses the right CTI concepts precisely and consistently to support the answer.

Task: `{task}`
Subtask: `{subtask}`
Prompt: `{prompt}`
Model Response: `{response}`
Return JSON only with this exact schema:

```
{
"writing_quality_score":  1,
"evidence_use_score":  1,
"cti_concept_focus_score":  1,
"notes":  "short explanation"
}
```

Rules: each score must be an integer from 1 to 4; use the rubric exactly; keep notes short and concrete; return JSON only.

Figure 9: Prompt template and rubric used for GPT-5.2 and Claude Sonnet 4.6 pointwise response-quality scoring.

Table 28: Agreement on the response-quality validation subset. QWK denotes quadratic weighted kappa.

| Comparison | QWK | Spearman |
|---|---|---|
| Annotator 1 vs. Annotator 2 | 0.6514 | 0.6461 |
| Annotator 1 vs. GPT-5.2 | 0.6019 | 0.7822 |
| Annotator 2 vs. GPT-5.2 | 0.5680 | 0.6440 |
| Annotator average vs. GPT-5.2 | 0.6313 | 0.7722 |
| Claude Sonnet 4.6 vs. GPT-5.2 | 0.7634 | 0.8266 |

Task-specific partial credit can be viewed as additional shaping, while the support issue studied here concerns whether the policy samples a fully verifier-correct answer. A policy $\pi_\theta(\cdot \mid x)$ therefore induces the answer-level success probability

$$p_\theta(a^\star \mid x) := \Pr_{y \sim \pi_\theta(\cdot \mid x)} [S(x, y; a^\star) = 1]. \tag{14}$$

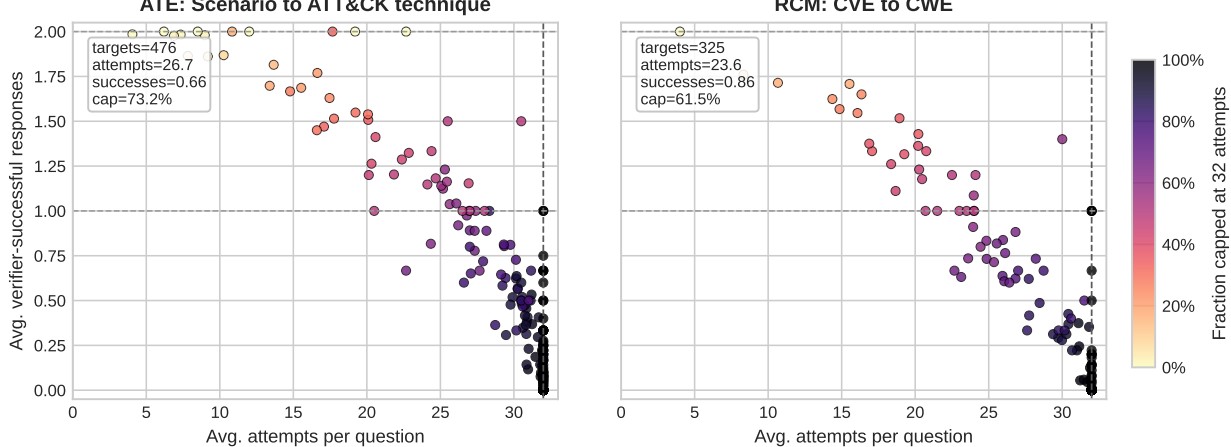

Figure 10: Target-level reward sparsity for ATE and RCM in DART-CTI stage-1 plain-generation attempts. Each point is one target identifier; the x-axis is mean attempts per question, the y-axis is mean verifier-successful responses per question, and color indicates the fraction of questions that reached the 32-attempt cap.

**Finite-budget detectability.** With a rollout budget of $k$ independent completions for the same prompt, the probability that the rollout group contains no fully verified completion is

$$\Pr[\text{no success in } k \text{ rollouts}] = (1 - p_\theta(a^\star \mid x))^k. \tag{15}$$

For a failure tolerance $\zeta \in (0, 1)$, define the exact detectability threshold

$$\varepsilon_{k,\zeta} := 1 - \zeta^{1/k} \approx \frac{-\log \zeta}{k}. \tag{16}$$

This is the minimum answer-level probability needed to make the chance of missing the correct answer in $k$ samples at most $\zeta$.

**Lemma J.1** (Finite-sample detectability). *Fix a prompt $x$, an answer $a \in \mathcal{A}$, and a rollout budget $k$. If $p_\theta(a \mid x) \geq \varepsilon_{k,\zeta}$, then*

$$\Pr[a \text{ is not observed in } k \text{ rollouts}] \leq \zeta. \tag{17}$$

*Equivalently, observing no successful rollout is a level-$\zeta$ event under any policy that assigns probability at least $\varepsilon_{k,\zeta}$ to the correct answer.*

*Proof.* If $p_\theta(a \mid x) \geq 1 - \zeta^{1/k}$, then

$$\Pr[a \text{ is not observed}] = (1 - p_\theta(a \mid x))^k \leq (\zeta^{1/k})^k = \zeta.$$

$\square$

**Small-budget all-zero regime.** The detectability threshold above characterizes when a success is reliably observable. A stronger "all-zero" regime occurs when the answer probability is so small that even one success is unlikely. For $\eta \in (0, 1)$, define

$$\delta_{k,\eta} := 1 - (1 - \eta)^{1/k} \approx \frac{\eta}{k}. \tag{18}$$

If $p_\theta(a^\star \mid x) \leq \delta_{k,\eta}$, then the probability of seeing no fully verified completion in $k$ rollouts is at least $1 - \eta$. Thus, when $p_\theta(a^\star \mid x) \ll 1/k$, the prompt can repeatedly produce all-zero rollout groups.

**Theorem J.2** (Small-budget support barrier for on-policy RLVR). *Fix a prompt $x$ and rollout budget $k$. Suppose that, at iteration $t$, $p_t := p_{\theta_t}(a^\star \mid x) \leq \delta_{k,\eta}$. Then, with probability at least $1 - \eta$, the $k$ on-policy rollouts for $x$ contain no fully verified completion. On this event, any per-prompt update rule whose direct positive signal for $a^\star$ comes only from observed verifier-correct rollouts receives no direct evidence for increasing the probability of $a^\star$ in that iteration.*

*Proof sketch.* The probability of no fully verified rollout is $(1 - p_t)^k$. Since $p_t \leq 1 - (1 - \eta)^{1/k}$, we have $(1 - p_t)^k \geq 1 - \eta$. Conditioning on this event, all sampled trajectories fail to reveal the correct answer, so an on-policy update that depends only on observed verified successes has no per-prompt positive example of $a^\star$ to reinforce. □

**MinervaRL as support seeding.** MinervaRL adds an auxiliary mechanism for prompts that fall into this sparse-support regime. When the base rollouts for a prompt $x$ contain no fully verified completion, MinervaRL constructs an answer-conditioned prompt

$$\tilde{x} = \mathrm{ACR}(x, a^\star, \mathrm{ref}(a^\star)),$$

which reveals the gold answer and optionally includes a truncated canonical reference. The EMA teacher samples ACR candidates from $\pi_\phi(\cdot \mid \tilde{x})$. A candidate is accepted only if it is verifier-correct under the original task target and passes the leakage and quality filters. The accepted trace is then distilled onto the original prompt $x$, not onto $\tilde{x}$.

We model this mechanism at the answer level through two finite-sampling assumptions. The first captures the probability that answer-conditioned generation yields an accepted verified trace; the second captures the effect of distilling that trace onto the original prompt. Under these assumptions, the theorem below characterizes when repeated accepted traces raise the correct-answer probability above the detectability threshold for a fixed rollout budget.

**Assumption J.3** (Answer-conditioned exposure). For a hard instance $(x, a^\star)$, before the prompt reaches the detectability threshold $\varepsilon_{k,\zeta}$, each ACR generation cycle has conditional probability at least $\alpha > 0$ of producing an accepted trace $y_{\mathrm{acr}}$ such that

$$S(x, y_{\mathrm{acr}}; a^\star) = 1. \tag{19}$$

**Assumption J.4** (Effective original-prompt distillation). Let $p_\theta(a^\star \mid x) > 0$. Whenever MinervaRL performs an SFT update on an accepted pair $(x, y_{\mathrm{acr}})$ with $S(x, y_{\mathrm{acr}}; a^\star) = 1$, the induced success probability under the original prompt increases by at least $\Delta > 0$ in log-space until the detectability threshold is reached:

$$\log p_{\theta^+}(a^\star \mid x) \geq \min\{\log \varepsilon_{k,\zeta}, \log p_\theta(a^\star \mid x) + \Delta\}. \tag{20}$$

**Theorem J.5** (Empirical support seeding via MinervaRL). *Fix a prompt-answer pair $(x, a^\star)$, rollout budget $k$, and failure tolerance $\zeta$. Let $\varepsilon_{k,\zeta} = 1 - \zeta^{1/k}$ and let $p_0 = p_{\theta_0}(a^\star \mid x) > 0$. Under Assumptions J.3 and J.4, MinervaRL raises $p_\theta(a^\star \mid x)$ to at least $\varepsilon_{k,\zeta}$ in a finite expected number of ACR-generation cycles. In particular, after*

$$L = \left\lceil \frac{[\log \varepsilon_{k,\zeta} - \log p_0]_+}{\Delta} \right\rceil \tag{21}$$

*successful original-prompt distillation updates, we have*

$$p_\theta(a^\star \mid x) \geq \varepsilon_{k,\zeta}. \tag{22}$$

*The expected number of ACR-generation cycles needed to obtain these $L$ accepted traces is at most $L/\alpha$. Consequently, once this threshold is reached, the probability that standard RLVR still misses the correct answer in $k$ rollouts is at most $\zeta$.*

*Proof sketch.* By Assumption J.4, each successful distillation update increases $\log p_\theta(a^\star \mid x)$ by at least $\Delta$ until the threshold $\varepsilon_{k,\zeta}$ is reached. Therefore $L$ accepted distillation updates suffice. By Assumption J.3, each ACR-generation cycle produces an accepted trace with conditional probability at least $\alpha$, so the expected number of cycles required to obtain $L$ accepted traces is at most $L/\alpha$. Finally, once $p_\theta(a^\star \mid x) \geq \varepsilon_{k,\zeta}$, Lemma J.1 implies that the probability of missing the correct answer in $k$ standard RLVR rollouts is at most $\zeta$. □

**Scope and limitations.** This argument is intentionally narrow. It explains how MinervaRL can reduce the incidence of zero-success rollout groups under a fixed, small rollout budget by seeding verified traces and distilling them onto the original prompt. It does not claim that ACR traces are faithful proofs, that every accepted trace improves general reasoning, or that answer conditioning is sufficient in domains where knowing the final answer does not help construct a valid derivation. It also does not analyze gradient interference between GRPO and SFT, partial-credit reward dynamics, checkpoint selection, or full language-model fine-tuning dynamics.

## K  Training Hyperparameters and Implementation Details

This appendix lists the training hyperparameters used for RLVR (GRPO), MinervaRL, and the controlled training baselines. We report only settings that directly affect optimization, sampling, or sequence truncation.

### K.1  RLVR (GRPO) settings

- **Optimizer:** GRPO with actor learning rate $1 \times 10^{-6}$.

- **Batching:** 128 prompts per training step.

- **Rollouts:** $N = 8$ sampled completions per prompt per step.

- **Sequence lengths:** max prompt length 2048 tokens; max response length 1024 tokens.

- **Schedule:** 500 training steps.

### K.2  MinervaRL (ACR + distillation) settings

- **Hard-example criterion:** mark a prompt "hard" if the best base-rollout reward is $< 1.0$ (CVSS prompts excluded).

- **ACR prompt context:** max ACR prompt length 4096 tokens; max response length 1024 tokens.

- **ACR sampling:** $K = 4$ traces per ACR prompt; temperature 0.7; nucleus sampling $p = 0.9$.

- **Deferred generation cadence:** generate/distill every $I = 10$ steps.

- **Teacher:** EMA teacher with decay $\alpha = 0.995$.

- **Distillation:** supervised fine-tuning on original prompts using up to 256 accepted traces per distillation interval; learning rate is scaled by $\gamma = 0.05$ relative to the RLVR learning rate.

- **Trace filtering:** a two-stage pipeline (heuristics + ML filter) is applied before distillation; the ML filter acceptance threshold is $\tau_q = 0.5$.

### K.3  Controlled baseline settings

**STaR-CTI.**

- **Trace generation:** maximum generation length 2048 tokens; temperature 0.7; top-$p = 0.95$.

- **Trace selection:** one original trace and one rationalization trace per training example in each round; verifier success threshold 1.0.

- **SFT training:** one epoch per round; train batch size 128; maximum sequence length 3072 tokens; bfloat16 FSDP; gradient checkpointing.

- **Optimizer:** AdamW with learning rate $1 \times 10^{-5}$, betas $(0.9, 0.95)$, weight decay 0.01, gradient clipping 1.0, and cosine scheduling with warmup ratio 0.1.

- **Micro-batching:** micro-batch size 8 for Llama-3.1-8B, Llama-3.2-3B, and Qwen3-8B; micro-batch size 4 for Qwen3-4B.

**DART-CTI.**

- **Corpus generation:** Llama-3.1-8B-Instruct teacher; maximum generation length 1024 tokens; temperature 0.7; top-$p = 0.95$; verifier threshold 1.0.

- **Rejection sampling:** plain stage targets two accepted traces per prompt with a cap of 32 attempts.

- **Guided fill:** maximum prompt length 4096 tokens; at most 8096 label-reference characters.

- **Filtering:** when enabled, use the response-only TextCNN filter with threshold 0.5, filter batch size 128, and filter maximum length 1024 tokens.

- **Student SFT training:** one epoch; train batch size 128; micro-batch size 8; maximum sequence length 4096 tokens; bfloat16 FSDP; gradient checkpointing.

- **Optimizer and selection:** AdamW with learning rate $1 \times 10^{-5}$, betas $(0.9, 0.95)$, weight decay 0.01, gradient clipping 1.0, and cosine scheduling with warmup ratio 0.1; checkpoints selected by synthetic-validation loss.

**LUFFY-CTI.**

- **Guidance corpus:** 32,000-row one-trace-per-prompt off-policy corpus derived from DART-CTI.

- **RL training:** GRPO advantages with actor learning rate $1 \times 10^{-6}$; train batch size 128; PPO mini-batch size 128; PPO micro-batch size 16; 500 training steps.

- **Rollout sampling:** $N = 8$ completions per prompt; maximum prompt length 2048 tokens; maximum response length 1024 tokens; rollout temperature 1.0; validation temperature 0.6.

- **Distributed training:** gradient checkpointing, dynamic batching, FSDP, maximum actor token length 32768 per GPU, and four GPUs per run.

- **Off-policy objective:** token-level off-policy loss, no off-policy normalization, `p_div_p_0.1` reshape setting, KL loss disabled, KL coefficient 0, entropy coefficient 0, and reference model disabled.

- **Prefix guidance:** one random prefix per prompt with maximum prefix length 1024 tokens and min/max prefix ratio 1.0.

### K.4 Timing overhead

To isolate per-step overhead, we ran a small-scale timing study on Llama-3.1-8B-Instruct for 10 training steps with validation disabled and one ACR/SFT interleave at step 10. Including end-to-end process time, GRPO took 469.8s and MinervaRL took 654.5s, a 39.3% increase. The measured MinervaRL components were: GRPO rollout 110.3s, GRPO reward 2.3s, ACR prompt construction 2.9s, ACR generation 36.4s, ACR reward 5.9s, and SFT distillation 9.3s. The remaining gap comes mainly from ACR log-prob computation (32.1s) and a modest increase in the shared actor update (+16.9s relative to GRPO). Thus, MinervaRL is not strictly Pareto-dominant at equal step count, but Figure 5 shows that the additional cost can be favorable in time-to-performance because MinervaRL reaches the best GRPO validation accuracy earlier and continues improving.

