# OpenReview forum: "Minerva: Reinforcement Learning with Verifiable Rewards for Cyber Threat Intelligence LLMs"
_TMLR — Under review for TMLR_

### Review · Reviewer_AKra · 2026-06-29

**Summary Of Contributions:**

The authors develop Minerva, a framework consisting of the Minerva-CTI dataset and the MinervaRL training algorithm, for cyber threat intelligence (CTI) tasks. The main focus is on mapping noisy, unstructured, or semi-structured security information into standardized CTI taxonomies, identifiers, and structured outputs. I appreciate the effort of the authors and think the topic is relevant. However, in its current form, the paper requires substantial additional work to be understandable and convincing to a broad ML audience

**Audience:**

Yes

**Audience Explanation:**

In my view, the results could be interesting to a broad group of researchers working on domain-specific LLM adaptation, especially in settings where neither standard GRPO nor standard SFT is sufficient. However, this potential contribution is currently weakened by the presentation, limited uncertainty analysis, and insufficiently clear ablations.

**Broader Impact Concerns:**

The work has potential dual-use implications, as also acknowledged in the authors’ Broader Impact Statement: improved CTI mapping and vulnerability reasoning can support defensive analysis, but may also help malicious actors organize threat information or prioritize attack-relevant vulnerabilities. The authors should briefly discuss concrete mitigation strategies, such as cyber-safety red-teaming, careful release of artifacts, and clearer documentation of intended defensive use cases.

**Claims And Evidence:**

No

**Claims Explanation:**

**Major Comments:**
- The current structure, terminology, evaluation protocol, notation, and presentation are not yet and the standard expected for TMLR or similar ML venues. In particular, the paper often assumes substantial CTI-specific background and introduces many domain-specific terms, abbreviations, and task names before making the basic ML contribution sufficiently clear.
- A major weakness of the empirical evaluation is the *lack of uncertainty quantification* and the *reliance on a heterogeneous aggregate score*. The paper reports broad results across many benchmarks, but many conclusions are drawn from point estimates and an “Avg.” column that averages fundamentally different metrics: multiple-choice accuracy, exact-match taxonomy prediction, multi-label F1, information-extraction F1, and normalized CVSS score. In my view, these quantities are not directly comparable and the aggregate average can obscure or skew the interpretation of the results.
- This is especially problematic because several reported improvements over strong baselines are small, while no confidence intervals, bootstrap intervals, standard errors, or multi-seed training variation are reported for the main results. Without uncertainty estimates, it is difficult to determine whether the claimed gains are statistically meaningful or within evaluation or training noise. At minimum, the authors should report confidence intervals over evaluation examples, paired significance tests or bootstrap tests for model comparisons, and multi-seed results for the key backbones and baselines.
- The absence of rollout-sensitive evaluation metrics is also surprising. The proposed method is motivated by sparse verifier feedback and finite rollout support: the central claim is that MinervaRL helps when a correct answer is unlikely to appear among a small number of sampled completions. While the paper reports useful training-dynamics plots such as the zero-solve fraction, the final benchmark evaluation does not appear to report whether MinervaRL actually improves multi-sample detectability on held-out tasks. This would directly test the paper’s claimed support-expansion mechanism. For exact-match identifier, multiple-choice, and taxonomy tasks, the authors should report pass@1 and pass@k, or verifier success@k, under matched sampling budgets. For multi-label, extraction, and graded-scoring tasks, they could report oracle best-of-k verifier score, best-of-k F1, or thresholded success@k. More generally, the paper should show the relationship between single-sample accuracy and best-of-k detectability across task families.
- Overall, the breadth of evaluation in terms of experiments and baselines is useful, but the current measurement protocol does not fully justify the strength of the aggregate claims. The authors should replace or supplement the heterogeneous average with task-family averages, per-task uncertainty intervals, sample-size-aware analyses, and rollout-aware metrics.
- Connected to the evaluation protocol, and related to the difficulty of understanding the contribution, is the MinervaRL algorithm itself. I think the idea developed by the authors is interesting and, given the results, potentially helpful for CTI. However, it is not fully clear which components help and by how much. The results indicate that only GRPO or only SFT are insufficient to reach the performance of the proposed algorithm. However, the contribution of the individual pipeline components remains unclear: the hard-prompt gate, the answer-conditioned rationale generation, the EMA teacher, the heuristic filters, the TextCNN classifier, and the final distillation step. The paper contains some ablations, but these are limited and do not fully explain which parts of the method are responsible for the main 12-task performance gains. I would like to see a clearer component-wise ablation on the main evaluation setting, including versions without ACR, without the EMA teacher, without the TextCNN filter, without heuristic filtering, and with matched additional SFT compute.
- Unfortunately, the theoretical sections do not match the standards of TMLR or similar venues. Let me take Section 4.1 as an example. The authors introduce GRPO, but several symbols and objects are not explained carefully enough. For example, $\theta$ is not introduced before use, and $\varepsilon$ and $\rho$ are not sufficiently described. For parameters, supports, and functions, the paper often does not specify the relevant domains and codomains. The distinction between $y$, $y^\star$, and $\hat{y}$ is also not always clear to me as a reader.
- I also find the Appendix H formalization of the support-expansion story weak. The appendix provides useful intuition, but the “proof” is largely a consequence of the assumptions. Lemma H.1 is a straightforward finite-sampling calculation. The main theorem then assumes that ACR generation produces accepted traces with probability at least α, and more importantly assumes that each distillation update increases the original-prompt success probability by at least a fixed $\Delta$ in log-space. This assumption is essentially the mechanism the theorem is supposed to justify. As a result, the theorem does not analyze GRPO, the interaction between GRPO and SFT, gradient interference across prompts, partial-credit rewards, or the actual optimization dynamics of language-model fine-tuning. I am not sure this theoretical section adds much beyond the already intuitive finite-sampling explanation, unless it is reframed more explicitly as a toy formalization rather than theoretical support for the full algorithm.

**Requested Changes:**

In addition to the major comments stated above, which need to be addressed, I have the following minor comments:

**Minor Comments:**
- I read the first two sentences multiple times before understanding that the paper is mainly about taking unstructured security data and classifying or mapping it into cyber intelligence frameworks. There are many terms such as “heterogeneous security artifacts,” “machine-readable representations,” and “triage” that are meaningful to CTI experts but not immediately clear to a broad ML audience. Since TMLR addresses a broad community, the introduction should explain the task more directly, ideally with a concrete example.
- The main idea of the paper was only clear to me after reading the paper multiple times. Figure 1 is a good example. To me, MinervaRL is an interesting combination of GRPO with periodic SFT, where the main invention is the pipeline for generating and filtering distilled samples. However, instead of highlighting the standard GRPO loop in one color and the additional ACR/SFT loop in another color, the figure presents a 15-step pipeline that is difficult to follow. A simpler figure separating the base GRPO update from the auxiliary distillation mechanism would make the contribution much clearer.
- Is $G$ introduced in Section 4.1 in Eq. (1) the same quantity as $N$ used later in Step 1 of Algorithm 1? If so, the notation should be unified. If not, the difference should be explained more clearly.
- The notation introduces $\mathcal{D}=\lbrace (x_i,y_i^\star \rbrace$, which is ambiguous: it appears to denote a set containing a single input and label rather than a dataset of paired examples. This should likely be written as $\mathcal{D}=\lbrace(x_i,y_i^\star)\rbrace_{i=1}^N$. This is another example of the insufficiently careful notation throughout the paper.
- The paper would benefit from a clearer explanation of the CTI tasks and metrics before the results table. At present, a reader must know many benchmark names and CTI taxonomies to interpret the performance numbers. A small table grouping tasks into multiple-choice QA, taxonomy mapping, multi-label prediction, CVSS scoring, and information extraction would make the evaluation much easier to understand.
- Several parts of the paper use strong aggregate language, such as “consistent gains” or “outperforms,” while the results are mixed across individual tasks and sometimes small for strong baselines. I suggest making these claims more cautious unless uncertainty estimates and task-level significance analyses are added.

---

> ### Author Response · Authors · 2026-07-10
>
> Thank you for the constructive review. We have revised the manuscript to incorporate the reviewers' feedback, with expanded uncertainty analysis, rollout-aware evaluation, component ablations, and clearer presentation of the CTI task and MinervaRL method. We have uploaded the revised manuscript, and the manuscript diff from the previous submission is included in the supplementary material. Below we address this reviewer's main suggestions and point to the corresponding changes in the revised paper.
>
> **Uncertainty and heterogeneous metrics.** We now supplement the aggregate performance results with task-level paired uncertainty. Revised Sec. 5.2 defines paired nonparametric bootstrap CIs over evaluation examples, with the main paired results in revised Tables 2-4: for each task/backbone/model comparison, we resample the same evaluation examples for MinervaRL and the baseline, recompute the original task metric, and report the percentile 95% CI for `MinervaRL - baseline`. Task metrics and denominators are in revised Table 15.
>
> Revised Table 2 reports per-task CIs for MinervaRL vs GRPO. Across 48 backbone-task comparisons, MinervaRL has positive point deltas on 34/48; the paired CI is entirely above zero on 28/48, below zero on 7/48, and overlaps zero on 13/48. Revised Table 3 extends this count summary to all baselines: MinervaRL has 35/48 CI-positive comparisons vs Base, 35/48 vs STaR, 34/48 vs DART, 28/48 vs GRPO, and 20/48 vs LUFFY; the corresponding point-win counts are 42/48, 38/48, 38/48, 34/48, and 32/48.
>
> Revised Table 4 further groups the same counts by task family: QA/selection (CKT, CyberMetric, SOCEval), taxonomy mapping (RCM, ATE, RMS, ElasticRule), vulnerability scoring (VSP), and information extraction (APTNER, LANCE, AnnoCTR, AZERG). Full per-task paired CIs for all baselines are in Appendix Tables 17-21, and the macro-average CI table is included only for comparability in Appendix Table 16. Multi-seed RLVR would further strengthen the study, but is not feasible for us given the compute needed to post-train these LLMs; we therefore qualify small-margin claims, including Qwen3-4B vs GRPO.
>
> **Rollout-aware evaluation.** We added rollout-aware best-of-\(k\) evaluation on the 12 CTI evaluation tasks under matched sampling budgets for GRPO and MinervaRL (revised Sec. 7.2 and Table 8). For each example, we sample eight responses from each model using temperature 0.7, top-\(p\) 0.95, and a 2048-token generation limit, then compute best-of-\(k\) for \(k\in\{1,2,4,8\}\). For exact-match, MCQ, and taxonomy tasks this is verifier success/best-of-\(k\) accuracy; for graded, extraction, and multi-label tasks we select the highest-scoring rollout per example and recompute the original task metric. Revised Table 8 summarizes the paired results: MinervaRL improves over GRPO on 32/48, 36/48, 38/48, and 39/48 comparisons for \(k=1,2,4,8\), respectively. At \(k=8\), 31/48 paired intervals are positive, 6/48 are negative, and 11/48 overlap zero. Full rollout scores and per-task rollout CIs are in Appendix Tables 22-26.
>
> **Component ablations.** We added final 12-task Llama-3.1-8B component controls in revised Sec. 6.3 and Table 5, with full per-task scores in Appendix Table 27. Full MinervaRL reaches 60.2 average, compared with 56.0 for GRPO/no ACR, 53.9 for GRPO with 12 rollouts, and 57.8 for in-loop answer-only SFT, which matches the auxiliary distillation schedule while replacing ACR traces with direct final-answer targets. Removing the EMA teacher, TextCNN filter, or filtering pipeline gives 57.2, 57.7, and 57.8, respectively. These results indicate that extra rollouts or direct answer-only SFT do not recover the full method, while the teacher/filtering components provide smaller but measurable contributions.
>
> **Presentation, notation, and theory.** We have added a concrete CVE-to-CWE/CVSS example (Figure 1) in the introduction, simplified the method diagram to separate the standard GRPO loop from the auxiliary ACR/SFT path (Figure 2), expanded the task/metric description before the results table, and cleaned up Sec. 4 notation by defining the verifier domain, dataset indexing, reward vectors, actor/teacher policies, and original versus answer-conditioned prompts. We also reframed the theory section and appendix as a finite-sampling formalization of the support-seeding intuition rather than a full optimization theory for GRPO plus SFT in language models; the appendix now states the assumptions and scope limitations explicitly.
>
> **Broader impact.** We expanded the broader-impact statement with concrete intended defensive uses, dual-use risks, reliability risks, and mitigations, including defensive-use documentation, model/data cards, cyber-safety review, and staged release when needed.

---

### Review · Reviewer_5cW6 · 2026-07-04

**Summary Of Contributions:**

This paper studies RLVR for CTI tasks, where many outputs are structured identifiers that can be verified deterministically. It introduces Minerva-CTI, a 16-task training suite with task-specific verifiers, and MinervaRL, a GRPO-based RLVR method that adds answer-conditioned rationale self-distillation for hard examples. The paper evaluates the method across four open-weight backbones and 12 CTI benchmarks, with several controlled baselines trained on the same data and verifiers.

The main strengths are clear. CTI is a natural domain for verifier-based RL, since many tasks have well-defined labels and evaluation rules. The paper is also unusually careful in documenting the training tasks, reward functions, filtering rules, and hyperparameters. The baseline comparisons are relatively fair, as most competing methods are re-implemented under the same data and verifier setting.

The main weaknesses are also important. First, the paper does not report variance or uncertainty, even though some of the reported margins are quite small. Second, checkpoint selection uses AthenaBench-Mini, while several final evaluation benchmarks also come from AthenaBench; the paper should clarify whether the instances are disjoint. Third, the theoretical appendix is fairly limited, since some key assumptions already encode the quantities that the experiments are meant to establish.

**Audience:**

Yes

**Audience Explanation:**

Two audiences will find this useful. First, the RLVR community: the paper is a carefully instrumented case study of extending verifiable-reward training beyond math/code to a domain with large, long-tailed canonical label spaces, and the sparse-reward failure mode it targets (all-zero rollout groups under small budgets) is a recognized, general problem; the hardness-gated answer-conditioned distillation recipe and its ablations are directly reusable. Second, the security-ML community: a 16-task verifier-checkable CTI training suite with documented reward functions, plus evidence that small open-weight models can be pushed well past larger security-SFT models on structured CTI tasks, is of clear practical interest (conditional on release — see requested changes). The negative/nuanced findings (answer-only SFT degrading OOD performance; judge-preference vs. verifier-score divergence) are also informative.

**Broader Impact Concerns:**

The broader impact statement should more directly discuss the dual-use implications of the results. In particular, CyberSecEval-2 CanaryExploit improves substantially after MinervaRL, and the paper presents this as a positive capability result. The revision should explain why improved exploit-generation performance is considered desirable, whether CTI-specialized training lowers the cost of offensive misuse, and how this affects the release plan for models and data.

**Claims And Evidence:**

Yes

**Claims Explanation:**

The main claims are generally supported by a broad and consistent set of experiments. In particular, MinervaRL improves over GRPO across all four backbones, and the zero-solve-fraction analysis gives a plausible explanation for why the proposed method helps. The paper also presents the results carefully, reporting average gains while acknowledging exceptions.

That said, I still see several weaknesses in the evidence. First, the paper does not report uncertainty estimates. Some important margins are quite small, so it is hard to know whether they are robust without multiple runs or confidence intervals. Second, the checkpoint selection protocol may partly overlap with the evaluation setting. Since AthenaBench-Mini covers several benchmark families used in the final evaluation, the paper should clarify whether the instances are disjoint and how this affects the transfer claims. Third, some evaluation details are missing, including the decoding setup and whether the same answer extraction rules are applied to all baselines.

Overall, the experimental evidence is reasonably strong, but these clarifications are needed to make the conclusions fully convincing. The theoretical appendix is useful as a formal description, but I would not treat it as independent evidence for the empirical claims.

**Requested Changes:**

The most important issue is the lack of uncertainty reporting. Since several key differences are small, the paper should add confidence intervals, at least by bootstrapping evaluation instances. Multiple training seeds for GRPO and MinervaRL on one representative backbone would further strengthen the main comparison. Claims based on margins around one point or less should either be statistically supported or stated more cautiously.

The checkpoint-selection protocol also needs clarification. AthenaBench-Mini is used for model selection, while several final evaluation benchmarks come from the same family. The paper should state whether the Mini instances are disjoint from the reported evaluation sets, confirm that all trained baselines use the same selection criterion, and discuss how this affects the transfer claim, especially for CKT.

The evaluation protocol should be made explicit. Please report the decoding setup for the main evaluations and clarify whether the same answer-extraction rules are applied to all systems, including external security-SFT baselines whose output formats may differ.

The paper should also add a data/code availability and licensing statement. Since Minerva-CTI is a central contribution, it is important to state whether the dataset, verifiers, and training code will be released, and how redistribution restrictions from the upstream CTI sources are handled.

Several additional experiments would further improve the paper. An in-loop answer-only distillation ablation would help separate the effect of rationales from simply injecting gold answers. A compute-matched GRPO baseline would address the extra cost of MinervaRL. A standard SFT-then-GRPO baseline would also be useful, since it is a common practical recipe. Finally, the theoretical appendix would be more convincing if its key assumptions were empirically checked, or if the theorem were framed more modestly in the main text.

---

> ### Author Response · Authors · 2026-07-10
>
> Thank you for the constructive review. We have revised the manuscript to incorporate the reviewers' feedback, with expanded uncertainty analysis, clearer checkpoint/evaluation protocol details, additional controls, and more concrete release and dual-use discussion. We have uploaded the revised manuscript, and the manuscript diff from the previous submission is included in the supplementary material. Below we address this reviewer's main suggestions and point to the corresponding changes in the revised paper.
>
> **Uncertainty and small margins.** We added paired nonparametric bootstrap CIs over evaluation examples in revised Sec. 5.2, with the main paired results in Tables 2-4 and full paired tables in Appendix Tables 16-21. Revised Table 2 reports per-task CIs for MinervaRL vs GRPO; across 48 backbone-task comparisons, MinervaRL has 34/48 positive point deltas, with 28 CI-positive, 7 CI-negative, and 13 overlapping-zero intervals. Revised Table 3 extends this to all baselines: CI-positive counts are 35/48 vs Base, 35/48 vs STaR, 34/48 vs DART, 28/48 vs GRPO, and 20/48 vs LUFFY. Appendix Table 16 reports macro-average CIs for comparability: vs GRPO, MinervaRL is +4.2 [3.5, 5.1], +6.2 [5.3, 7.1], +6.4 [5.6, 7.6], and +0.4 [-0.8, 1.3] pp for Llama-8B, Llama-3B, Qwen-8B, and Qwen-4B. We therefore qualify small-margin cases, especially Qwen3-4B vs GRPO and Qwen3-8B vs LUFFY. Multi-seed RLVR would further strengthen the study, but is not feasible for us given the compute needed to post-train these LLMs.
>
> **Checkpoint selection and AthenaBench-Mini.** Revised Sec. 5.2 now states that GRPO, MinervaRL, and the controlled trained baselines use the same checkpoint-selection criterion: average Minerva-Dev and AthenaBench-Mini performance. To directly answer the disjointness question: AthenaBench-Mini is not fully instance-disjoint from the Athena-derived final evaluation columns; it is a subset of the CKT, RCM, VSP, ATE, and RMS tasks. However, the comparison remains symmetric because all trained methods use the same selection rule. As a sensitivity check, we removed the Mini instances and recomputed MinervaRL-GRPO on the remaining examples for those five columns. The five-task non-Mini deltas are +7.7 [6.4, 9.0], +11.9 [10.8, 13.1], +8.6 [7.6, 9.7], and -1.3 [-2.1, -0.5] pp for Llama-8B, Llama-3B, Qwen-8B, and Qwen-4B, respectively; this supports the Llama/Qwen-8B conclusions and is consistent with our softened Qwen3-4B claims.
>
> **Evaluation protocol.** Revised Sec. 5.2 now specifies the main decoding and scoring setup: the main single-sample results use greedy decoding with temperature 0.0 and a 2048-token generation limit; every system is evaluated with the same fixed benchmark prompts, task-specific parsers/normalizers, and scoring scripts; and unparseable outputs are treated as empty predictions. These rules are shared across all systems, including external security-SFT baselines. The separate rollout-aware analysis in revised Sec. 7.2/Table 8 uses sampled decoding with temperature 0.7, top-\(p\) 0.95, and matched eight-response budgets.
>
> **Additional ablations.** Following the reviewer's suggestion, we added a new in-loop answer-only distillation ablation and report final 12-task Llama-3.1-8B ablations in revised Sec. 6.3/Table 5, with full per-task scores in Appendix Table 27. Full MinervaRL reaches 60.2 average, compared with 56.0 for GRPO/no ACR, 53.9 for GRPO with 12 rollouts, 57.8 for in-loop answer-only SFT, 57.2 without the EMA teacher, 57.7 without the TextCNN filter, and 57.8 without filtering. The in-loop answer-only control directly tests whether gains come from simply injecting gold answers, while GRPO-12 controls additional rollout sampling; neither recovers the full method.
>
> **Theory, release, and dual-use.** We reframed the theory section and appendix as a finite-sampling formalization of the support-seeding intuition, not independent evidence for full GRPO+SFT language-model optimization; the appendix now states the assumptions and scope limitations explicitly. We also added a Data and Code Availability statement: we will release the Minerva codebase, data-construction pipeline, evaluation tools, and derived Minerva-CTI splits for research use while preserving upstream notices, licenses, and attribution. Finally, we expanded the broader-impact statement to discuss defensive use, reliability risks, improved CanaryExploit performance as a dual-use signal rather than a deployment objective, and mitigations including defensive-use documentation, model/data cards, cyber-safety review, and staged release when needed.

---

### Review · Reviewer_T8JQ · 2026-07-07

**Summary Of Contributions:**

## Summary

This paper introduces **Minerva**, a verifiable-reward RL framework for cyber threat intelligence (CTI). Its key insight is that many CTI tasks, such as ATT&CK mapping, CWE prediction, CVSS scoring, and mitigation selection, have canonical answers or fixed schemas, making them well suited for programmatic verification and RLVR. The authors build **Minerva-CTI**, covering 16 verifiable CTI task types, and propose **MinervaRL** to address the sparse-reward issue in standard GRPO. For hard samples, MinervaRL uses gold labels to generate answer-conditioned rationales, filters them through verification, and distills the resulting traces back into the original answer-free prompts. Experiments across 12 CTI benchmarks and 4 open-weight backbones show consistent gains, especially on structured verifiable tasks. Overall, I think the paper convincingly shows that CTI is a strong application domain for verifier-driven post-training, and that ACR is a useful mechanism for reducing sparse rewards in RLVR.


## Strengths

1. The paper studies a meaningful and well-motivated problem in CTI. Many CTI tasks require standardized identifiers or structured schemas, which naturally have canonical answers and can be automatically verified.

2. The proposed task suite is broad and fairly complete. Minerva-CTI covers 16 types of verifiable CTI tasks, so the paper is not limited to a single narrow benchmark. Instead, it attempts to build a more unified verifier-based training suite for CTI.

3. The experimental evaluation is comprehensive. The authors evaluate on 12 CTI benchmarks and 4 open-weight backbone models, and compare against base models, GRPO, STaR-CTI, DART-CTI, LUFFY-CTI, and external security-SFT models.

4. The ablation study is convincing. The paper compares several variants, including more GRPO rollouts, answer-only SFT, removing the EMA teacher, removing verification filtering, and removing the ML filter. These results suggest that the gains come from the full MinervaRL pipeline rather than one isolated component.

5. The analysis is stronger than simple benchmark reporting. The paper provides theoretical insights and uses the zero-solve fraction to show that MinervaRL reduces the number of samples where all rollouts fail. This directly supports the sparse-reward motivation.

6. The method is especially well matched to structured CTI tasks such as CWE prediction, CVSS vector prediction, ATT&CK technique mapping, and mitigation recommendation, where automatic verification is both practical and meaningful.

**Audience:**

Yes

**Audience Explanation:**

I believe this paper would be interesting to at least part of the TMLR audience.

- Researchers working on RLVR and verifier-based post-training may find this paper useful because it provides a natural and practically meaningful application domain.

- Researchers interested in domain-specific LLM post-training may also find the paper relevant, since it shows how domain knowledge, structured outputs, and automatic verification can be combined in a systematic way.

- Readers working on cybersecurity or CTI may find the paper valuable because it introduces a broad verifiable CTI task suite and demonstrates the potential of RL-style training on structured security tasks.

That said, the paper may be less appealing to readers looking for a general-purpose RL algorithmic contribution. The main advantage of MinervaRL is closely tied to structured and verifiable CTI tasks, so I view it more as a domain-driven RLVR framework than a fully general RL method.

**Broader Impact Concerns:**

I do not see major ethical concerns that would directly prevent publication. However, since CTI is inherently dual-use, the paper should discuss broader impact more explicitly.

- On the positive side, stronger CTI models could help security analysts perform vulnerability mapping, threat classification, and mitigation recommendation more efficiently.

- On the risk side, stronger CTI models could also be useful to malicious actors by improving vulnerability understanding, attack technique mapping, or automated threat analysis.

- The authors should clearly state the intended defensive use cases of the proposed method and discuss possible misuse risks.

- The authors should also discuss reliability risks. In CTI settings, incorrect CWE labels, ATT&CK techniques, CVSS vectors, or mitigation recommendations could mislead analysts or affect downstream security decisions.

Overall, I suggest that the authors add a broader impact statement, especially around the dual-use nature of CTI, model reliability, and boundaries for defensive use.

**Claims And Evidence:**

Yes

**Claims Explanation:**

Overall, the main claims of the paper are mostly supported by the experimental evidence.

- The claim that CTI is suitable for verifier-driven post-training is convincing. Tasks such as ATT&CK technique prediction, CWE prediction, CVSS vector prediction, and mitigation selection have standardized outputs and clear verification rules, making them suitable for programmatic verifiers.

- The claim that MinervaRL improves over standard GRPO is supported by the reported results. The paper shows consistent improvements across 12 CTI benchmarks and 4 open-weight backbone models, with an average gain of 15.8 percentage points over the base model and 4.3 percentage points over standard GRPO.

- The sparse-reward motivation is also directly supported. The zero-solve fraction analysis shows that MinervaRL reduces the proportion of rollout groups where all completions fail, which is consistent with the motivation behind ACR.

- The ablation study supports the importance of the full pipeline. The performance drops observed when removing the EMA teacher, filtering, ML filter, or distillation suggest that these components all contribute to the final performance.

However, I think some claims should be stated more cautiously.

- The generalization claim should be softened. The strongest evidence comes from training-aligned, structured, verifier-friendly tasks, while the gains on not-in-training tasks are smaller.

- The claim about improved reasoning should also be made more carefully. Since ACR generation uses gold labels, the current experiments do not fully rule out the possibility that the model learns answer-conditioned rationale patterns rather than genuinely improving CTI reasoning.

- The claim about real-world CTI usefulness needs more analysis. Verifier scores can evaluate whether structured answers are correct, but they do not fully capture analyst-facing CTI quality.

**Requested Changes:**

- The authors should more thoroughly discuss the risk of shortcut learning or reward hacking introduced by gold-label-conditioned rationales. Since ACR traces are generated with access to the answer, the paper should explain why this mechanism does not simply teach the model to imitate answer-driven explanations, but instead improves CTI reasoning.

- The authors should provide a clearer cost-performance analysis. MinervaRL introduces 39.3% additional training overhead compared with GRPO, while the average improvement over GRPO is 4.3 percentage points. The paper should clarify in which scenarios this extra cost is justified.

- The authors should weaken or more precisely state their generalization claims. The current results show that MinervaRL performs best on training-aligned tasks, while improvements on not-in-training tasks are more limited. The paper should explicitly acknowledge that the method is mainly effective for verifier-aligned structured tasks.

- The authors should discuss the limitations of verifier-based evaluation more thoroughly. Real CTI analysis is not only about producing the correct standardized answer; it also requires sufficient evidence, trustworthy explanations, and technically correct reasoning.

### Changes That Would Strengthen the Paper

- The paper would benefit from a deeper analysis of the reliability of the filtering mechanism. The current lightweight heuristics and TextCNN filter may not be sufficient to evaluate semantic quality, evidence grounding, and technical correctness in complex CTI rationales.

- The paper could include a more fine-grained failure-mode analysis of hard prompts. The current definition of “no fully verified rollout” mixes formatting errors, missing knowledge, reasoning failures, and ambiguous samples, which may require different training strategies.

- The authors could discuss how to better use partial rewards or partial correctness signals. Many CTI tasks may have partially correct answers, while the current MinervaRL design mainly relies on fully verified completions for hard-prompt gating.

- The paper could include more analyst-facing evaluation. Metrics such as prose preference, evidence grounding quality, and explanation correctness would provide a more complete picture of model quality in realistic CTI use cases.

---

> ### Author Response · Authors · 2026-07-10
>
> Thank you for the constructive review. We have revised the manuscript to incorporate the reviewers' feedback, with expanded uncertainty analysis, clearer scope statements, additional ablations, and a more concrete broader-impact discussion. We have uploaded the revised manuscript, and the manuscript diff from the previous submission is included in the supplementary material. Below we address this reviewer's main suggestions and point to the corresponding changes in the revised paper.
>
> **Gold-label-conditioned ACR and shortcut learning.** We agree that ACR should not be interpreted as proof of faithful reasoning. Revised Sec. 4.2 now states that the gold label is used only to generate candidate training traces; accepted traces are verified/filtered and distilled back onto the original answer-free prompt; and all evaluations use the same answer-free prompts as GRPO. We also added a targeted Llama-3.1-8B ablation in revised Sec. 6.3/Table 5, with full per-task scores in Appendix Table 27. Full MinervaRL reaches 60.2 average, compared with 56.0 for GRPO/no ACR and 57.8 for in-loop answer-only SFT, which matches the auxiliary distillation schedule while replacing ACR traces with direct final-answer targets. This does not prove mechanistic reasoning faithfulness, but it shows that the gain is not recovered by simply injecting gold final answers under the same in-loop update schedule.
>
> **Cost-performance tradeoff.** Revised Sec. 8 discusses the training overhead directly. MinervaRL adds ACR generation, log-probability computation, and periodic distillation; Appendix K.4 reports a 39.3% increase in a 10-step Llama-3.1-8B timing study. This is a training-time cost, not an inference-time cost. The tradeoff is most justified for verifier-aligned CTI settings where a one-time post-training cost is acceptable to improve deployed structured-output accuracy. Revised Fig. 5 also shows that MinervaRL reaches strong validation performance faster than GRPO, so the overhead should be interpreted together with training dynamics rather than only per-step cost.
>
> **Generalization scope.** We softened the claims to reflect that MinervaRL is strongest on verifier-aligned structured tasks. Revised Sec. 6.5 and Tables 6-7 split evaluation tasks into training-aligned and not-in-training groups. MinervaRL gives the largest gains on aligned tasks, while not-in-training gains are broader relative to base models and more mixed relative to GRPO. The conclusion now states this more narrowly rather than implying broad reasoning improvement.
>
> **Verifier limits and analyst-facing quality.** We agree that verifier scores alone do not capture all CTI usefulness. The paper already includes an analyst-facing response-quality evaluation in revised Sec. 6.4/Fig. 3, with the judging rubric covering writing quality, prompt-evidence use, and CTI concept use. Appendix I includes the pointwise prompt/rubric and Table 28 validates GPT-5.2 judgments against human annotators and Claude Sonnet 4.6. We also expanded the broader-impact statement to say that outputs should be analyst-assistive rather than authoritative because incorrect CWE, ATT&CK, CVSS, or mitigation outputs can mislead downstream security decisions. We still view explanation faithfulness and deeper evidence-grounding evaluation as important future work, but the current revision separates verifier correctness from judged response quality.
>
> **Partial rewards and hard-prompt failures.** MinervaRL does not discard partial rewards in GRPO: revised Sec. 4.1 and Appendix B describe exact matching, structured partial credit for hierarchical labels, set overlap for multi-label targets, and dense CVSS scoring. The hard-prompt gate intentionally uses the stricter condition `max reward < 1` because ACR is meant to seed fully verified trajectories when no fully correct rollout is observed. We do not differentiate failure causes during training, since both GRPO and MinervaRL consume verifier rewards rather than manual error labels; separating missing knowledge, incorrect mapping, ambiguity, and formatting failures is useful future work.
>
> **Filtering reliability and broader impact.** Appendix D reports the TextCNN filter construction and diagnostics, including acceptance dynamics (Fig. 7), confusion matrix (Table 12), and false-positive/false-negative breakdowns (Tables 13-14). Revised Sec. 8 also states that stronger LLM-based filters may improve semantic quality but would increase cost. Finally, we expanded the broader-impact statement with intended defensive uses, misuse risks, reliability risks, and mitigations including defensive-use documentation, model/data cards, cyber-safety review, and staged release when needed.

---

### Review · Reviewer_9rjT · 2026-07-07

**Summary Of Contributions:**

This paper presents Minerva, an RLVR framework for Cyber Threat Intelligence (CTI). The authors first construct Minerva-CTI, a training suite consisting of verifier-checkable CTI tasks, including vulnerability classification, ATT&CK mapping, CVSS prediction, and mitigation recommendation. Building on this benchmark, they propose MinervaRL, which extends GRPO with hardness-gated answer-conditioned rationale (ACR) generation and periodic distillation to alleviate sparse verifier rewards. The proposed method is evaluated on four open-weight LLMs and twelve CTI benchmarks, showing consistent improvements over GRPO and several CTI-specific baselines. The paper also includes ablation studies, training dynamics analysis, and a preliminary transfer experiment on text-to-SQL.

The paper addresses an important application domain where verifier-based reinforcement learning is particularly well suited, and the benchmark itself is likely to be useful for future research. The empirical evaluation is broad and carefully conducted. However, I found the methodological contribution to be relatively incremental, and several aspects of the experimental methodology make it difficult to fully support the broader claims regarding reasoning improvement and generalization.

**Additional Comments:**

Overall, I think this is a solid empirical study with a valuable benchmark contribution. The experimental evaluation is extensive, and the benchmark itself is likely to become a useful resource for future research in verifier-based CTI training.

My main reservation concerns the methodological contribution. MinervaRL largely combines existing components—including GRPO, answer-conditioned rationale generation, self-distillation, and periodic supervised updates—into a practical pipeline. While this integration is well executed, I am not convinced that it constitutes a sufficiently strong algorithmic contribution for TMLR without stronger empirical evidence and more rigorous evaluation.

I encourage the authors to strengthen the empirical methodology, clarify the evaluation protocol, and moderate several claims regarding reasoning and generalization. Addressing these issues would substantially improve the paper.

**Audience:**

Yes

**Audience Explanation:**

I believe this work will be of interest to several communities within TMLR.

Researchers working on reinforcement learning with verifiable rewards may find this paper valuable because it explores an application domain that naturally supports deterministic verification through standardized taxonomies and identifiers. The proposed benchmark also extends RLVR beyond commonly studied domains such as mathematics and code generation.

The paper is also likely to interest researchers in cybersecurity and domain-specific LLM adaptation. Minerva-CTI provides a relatively comprehensive collection of verifier-checkable CTI tasks, and the experimental results demonstrate that RL-based post-training can substantially improve structured CTI prediction tasks.

That said, readers primarily interested in algorithmic advances in reinforcement learning may find the methodological novelty somewhat limited, since MinervaRL mainly combines existing techniques rather than introducing a fundamentally new optimization method.

**Broader Impact Concerns:**

The paper includes a Broader Impact Statement, and I appreciate that it acknowledges the dual-use nature of cyber threat intelligence. However, I believe this discussion could be expanded.

In particular, the paper demonstrates improved capability on structured CTI tasks that may also facilitate offensive analysis, including vulnerability understanding and attack technique mapping. The authors should discuss how these potential misuse risks influence their plans for releasing models, datasets, and training artifacts.

In addition, the paper could discuss reliability risks more explicitly. Incorrect predictions of CWE labels, ATT&CK techniques, CVSS vectors, or mitigation recommendations may influence downstream security analysis, and verifier-based evaluation alone does not fully capture the quality of analyst-facing explanations.

**Claims And Evidence:**

No

**Claims Explanation:**

The empirical results generally support the claim that MinervaRL improves performance over standard GRPO on the evaluated CTI benchmarks. The experimental section is comprehensive, covering multiple backbone models, controlled baselines, ablation studies, and analyses of training dynamics. The reduction in zero-solve fraction also provides a plausible explanation for why the proposed method is effective under sparse verifier rewards.

However, I do not think the current evidence fully supports several of the broader conclusions made in the paper.

First, the paper repeatedly suggests that MinervaRL improves reasoning. Since the rationale generator is explicitly conditioned on the gold answer, the current experiments cannot distinguish between genuine reasoning improvements, answer-conditioned explanation imitation, and simply learning from additional supervised signals. An answer-only distillation baseline would be necessary to isolate the contribution of rationale distillation.

Second, the paper makes relatively broad generalization claims. While MinervaRL performs well on tasks aligned with the training objectives, the improvements on tasks outside the training objectives are considerably smaller, and the evidence beyond CTI consists of only a single text-to-SQL experiment. I believe these results are encouraging, but not sufficient to demonstrate broad applicability.

Finally, the paper reports only point estimates. No confidence intervals, bootstrap analysis, statistical significance tests, or multi-seed experiments are provided. Since several reported improvements over GRPO are relatively modest, uncertainty estimates would be important for determining the robustness of the conclusions.

**Requested Changes:**

The following issues should be addressed before I would be comfortable recommending acceptance.
Critical
1.Report uncertainty estimates. The paper should include confidence intervals, bootstrap evaluation, or multi-seed experiments for the main comparisons. This is particularly important because several improvements over GRPO are relatively small.
2.Clarify the contribution of answer-conditioned rationale distillation. The current experiments do not distinguish between improved reasoning and additional supervision from gold-answer-conditioned rationales. An answer-only distillation baseline, or another control experiment that isolates the effect of rationale generation, would significantly strengthen the paper.
3.Moderate the generalization claims. The strongest improvements are observed on training-aligned tasks, while gains on tasks outside the training objectives are more limited. The discussion should more clearly reflect this distinction and avoid suggesting broad reasoning improvements that are not directly supported by the experiments.
4.Improve evaluation transparency. The paper should clarify the checkpoint-selection protocol, describe whether AthenaBench-Mini overlaps with downstream evaluation datasets, and provide additional details regarding decoding settings, answer extraction, and evaluation procedures.
Would strengthen the paper
1.Provide compute-matched comparisons against stronger GRPO baselines or discuss the trade-off between the reported performance gains and the approximately 39% additional training cost.
2.Report task-family averages instead of relying primarily on a single aggregate average across heterogeneous evaluation metrics.
Reframe the theoretical analysis as an intuitive explanation rather than a formal justification, since several assumptions already encode the desired outcome.
3.Reframe the theoretical analysis as an intuitive explanation rather than a formal justification, since several assumptions already encode the desired outcome.

---

> ### Author Response · Authors · 2026-07-10
>
> Thank you for the constructive review. We have revised the manuscript to incorporate the reviewers' feedback, with expanded uncertainty analysis, additional ablations, more precise claim scope, clearer evaluation protocol, and a stronger broader-impact/release discussion. We have uploaded the revised manuscript, and the manuscript diff from the previous submission is included in the supplementary material. Below we address this reviewer's main suggestions and point to the corresponding changes in the revised paper.
>
> **Uncertainty estimates.** We added paired nonparametric bootstrap CIs over evaluation examples in revised Sec. 5.2, with main results in revised Tables 2-4 and full paired tables in Appendix Tables 16-21. Revised Table 2 reports per-task CIs for MinervaRL vs GRPO: across 48 backbone-task comparisons, MinervaRL has 34/48 positive point deltas, with 28 CI-positive, 7 CI-negative, and 13 overlapping-zero intervals. Revised Table 3 extends this across baselines: CI-positive counts are 35/48 vs Base, 35/48 vs STaR, 34/48 vs DART, 28/48 vs GRPO, and 20/48 vs LUFFY. Appendix Table 16 reports macro-average CIs for comparability; vs GRPO, MinervaRL is +4.2 [3.5, 5.1], +6.2 [5.3, 7.1], +6.4 [5.6, 7.6], and +0.4 [-0.8, 1.3] pp for Llama-8B, Llama-3B, Qwen-8B, and Qwen-4B. We therefore qualify small-margin cases, especially Qwen3-4B vs GRPO and Qwen3-8B vs LUFFY.
>
> **Answer-conditioned rationales and compute controls.** We added one new control that directly targets this concern: an in-loop answer-only SFT ablation, which matches the auxiliary distillation schedule but replaces ACR traces with direct final-answer targets. We report it with the final 12-task Llama-3.1-8B ablations in revised Sec. 6.3/Table 5, with full per-task scores in Appendix Table 27. Full MinervaRL reaches 60.2 average, compared with 57.8 for in-loop answer-only SFT and 56.0 for GRPO/no ACR; thus direct answer-only distillation improves over GRPO but does not recover the full ACR method. Table 5 also reports other controls, including no EMA teacher, no TextCNN filter, no filtering, and the existing GRPO-12 sampling control. Revised Sec. 4.2 clarifies that gold labels are used only to generate candidate training traces; all evaluations use original answer-free prompts.
>
> **Generalization and task-family scope.** We moderated generalization and reasoning claims. Revised Sec. 6.5 and Tables 6-7 split the 12 CTI evaluation tasks into training-aligned and not-in-training groups. The largest gains are on aligned verifier-structured tasks, while not-in-training gains are broader relative to base models and more mixed relative to GRPO. Revised Table 4 also summarizes results by task family rather than relying only on the aggregate average. The conclusion now frames MinervaRL as most effective for verifier-aligned structured CTI tasks and avoids claiming broad reasoning improvement.
>
> **Evaluation transparency and checkpoint selection.** Revised Sec. 5.2 states that GRPO, MinervaRL, and controlled trained baselines use the same checkpoint-selection criterion: average Minerva-Dev and AthenaBench-Mini performance. AthenaBench-Mini is not fully instance-disjoint from Athena-derived final columns; it is a subset of CKT, RCM, VSP, ATE, and RMS. As a sensitivity check, after removing Mini instances from those five columns, MinervaRL-GRPO remains +7.7 [6.4, 9.0], +11.9 [10.8, 13.1], and +8.6 [7.6, 9.7] pp for Llama-8B, Llama-3B, and Qwen-8B, while Qwen-4B is -1.3 [-2.1, -0.5], consistent with our softened Qwen3-4B claims. We also added decoding/scoring details in revised Sec. 5.2: main single-sample results use greedy decoding with temperature 0.0 and a 2048-token generation limit, and all systems, including external security-SFT baselines, use the same prompts, parsers/normalizers, and scoring scripts.
>
> **Theory, cost, and broader impact.** We reframed the theory section and appendix as a finite-sampling formalization of the support-seeding intuition, not independent evidence for full GRPO+SFT language-model optimization; the appendix now states the assumptions and scope limitations explicitly. Revised Sec. 8 and Appendix K.4 discuss the 39.3% training overhead; this is a training-time cost, not an inference-time cost. Finally, we expanded data/code availability and broader impact: we will release the Minerva codebase, data-construction pipeline, evaluation tools, and derived splits for research use while preserving upstream notices/licenses, and we discuss defensive use, misuse risks, analyst-facing reliability risks, cyber-safety review, model/data cards, and staged release when needed.